# PHYX: DOES YOUR MODEL HAVE THE "WITS" FOR PHYSICAL REASONING?

## ABSTRACT

Existing benchmarks fail to capture a crucial aspect of intelligence: *physical reasoning*, the integrated ability to combine domain knowledge, symbolic reasoning, and understanding of real-world constraints. To address this gap, we introduce PHYX: the first large-scale benchmark designed to assess models' capacity for physics-grounded reasoning in visual scenarios. PHYX includes 3K meticulously curated multimodal questions spanning **6** reasoning types across **25** sub-domains and **6** core physics domains: thermodynamics, electromagnetism, mechanics, modern physics, optics, and wave & acoustics. In our comprehensive evaluation, even state-of-the-art models struggle significantly with physical reasoning. **GPT-o4-mini**, **Gemini-2.5-Pro**, and **GPT-5** achieve only **45.8%**, **62.4%**, and **65.2%** accuracy respectively—performance gaps exceeding **10%** compared to human experts. Our analysis exposes critical limitations in current models: *over-reliance on memorized disciplinary knowledge*, *excessive dependence on mathematical formulations*, and *surface-level visual pattern matching* rather than genuine physical understanding. We provide in-depth analysis through fine-grained statistics, detailed case studies, and multiple evaluation paradigms to thoroughly examine physical reasoning capabilities. To ensure reproducibility, we implement an evaluation protocol based on widely-used toolkits such as VLMEvalKit and lmms-eval, enabling one-click evaluation. All source code and data are available on our anonymous repository: anonymous.4open.science.

## 1 INTRODUCTION

*Physics is the most fundamental and all-inclusive of the sciences.*

– Richard Feynman

State-of-the-art models (Guo et al., 2025; OpenAI, 2024b; Team, 2025) can now basically solve Olympiad-level mathematical problems with human-competitive accuracy on benchmarks including AIME (MAA, 2024), GPQA (Rein et al., 2024), MATH-500 (Hendrycks et al., 2021), Olympiad-Bench (He et al., 2024), *etc.* Emerging multimodal large language models (MLLMs) like GPT-4o (OpenAI, 2024a) further offer promising pathways by combining visual understanding with reasoning capabilities. Recent advances in multimodal foundation models have spurred the development of benchmarks assessing disciplinary knowledge (Yue et al., 2024) and mathematical problems (Wang et al., 2024a; Zhang et al., 2024; Lu et al.). However, these evaluations overlook a critical dimension of machine intelligence: *physical reasoning*, the ability to integrate disciplinary knowledge, symbolic operations, and understanding of real-world constraints.

To address these gaps, we present **PHYX**, the first large-scale benchmark designed for evaluating physics-based reasoning via multimodal problem-solving with three core innovations: (1) 3,000 newly collected questions with realistic physical scenarios requiring integrated visual analysis and causal reasoning, (2) Expert-validated data design covering six fundamental physics domains with representative examples illustrated in Figure 2, and six distinct physical reasoning types, (3) Strict unified three-step evaluation protocols account for varying instruction-following capabilities across models and enable accurate assessment of reasoning. Each scenario undergoes rigorous validation by physics Ph.D. students to ensure scientific accuracy while eliminating dataset bias.

In addition to MLLMs, our benchmark supports evaluating LLMs by translating the images into text descriptions, thereby enabling an assessment of LLMs on these visually-grounded tasks. Our evalua-

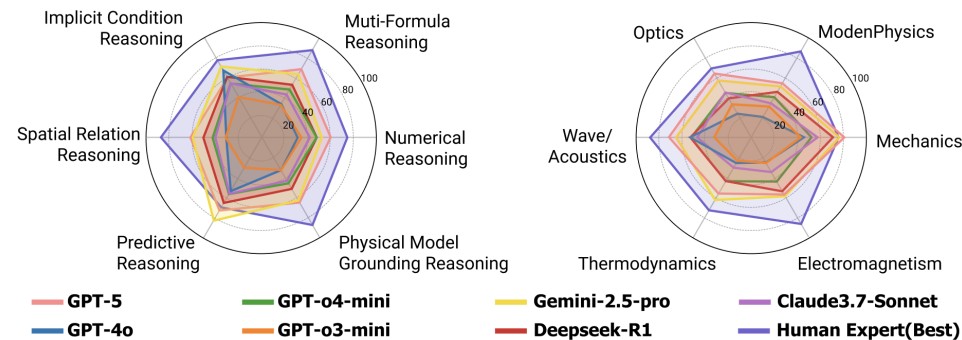

Figure 1: Accuracies of three leading MLLMs, two leading LLM and human performance on our proposed PHYX across 6 physical reasoning types and 6 domains.

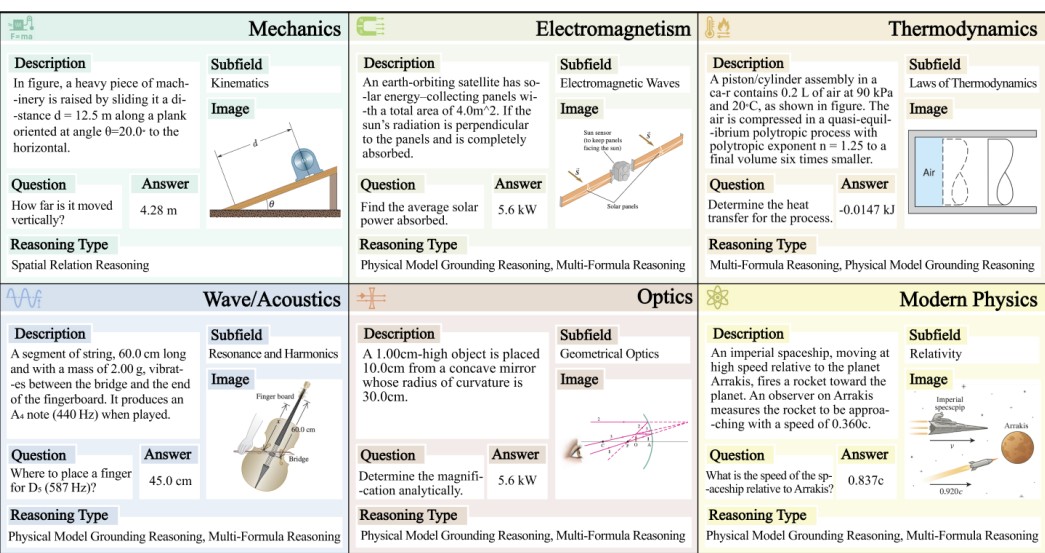

Figure 2: Sampled PHYX examples from each domain.

tion of 16 foundation models reveals an unprecedented capability gap: While the worst-performance group of physics undergraduates and graduates achieves 75.6% accuracy, the best-performing MLLM (GPT-5) scores only 65.2%. This 10-point performance chasm persists across all physics domains, most notably in Modern Physics (human 86.7% vs. model 56.5%) and Wave/Acoustics (human 86.7% vs. model 71.0%), as shown in Figure 1.

These results expose three critical shortcomings in current multimodal reasoning frameworks: (1) Visual reasoning errors (39.6%) indicate that models frequently misinterpret visual context, underscoring their limited capability in extracting and reasoning from physical scenarios. (2) The inconsistent performance across input variations: Full-Text, Text-DeRedundancy, and Text-Minimal, demonstrates that MLLMs remain overly dependent on textual descriptions, failing to effectively leverage visual input for reasoning. (3) Comparing physical reasoning performance to mathematical reasoning benchmarks such as MathVerse (Lu et al.) and MATH-V (Wang et al., 2024a) reveals that physical reasoning poses significantly greater challenges, highlighting a critical need for improved integration of abstract concepts and real-world knowledge. PHYX thus provides both a toolkit for model improvement and a roadmap for developing physically-grounded AI systems.

Our contributions can be summarized as follows: **Novel Benchmark Design:** We introduce PHYX, the first large-scale benchmark for evaluating the reasoning capabilities in the physical world for both multi-modal models and language models. Curated by experts, it spans 25 fine-grained domains and 6 reasoning types with realistic scenarios. **Versatile Evaluation Framework:** PHYX supports versatile evaluation frameworks, including *assessment formats* (multiple-choice vs. open-ended)

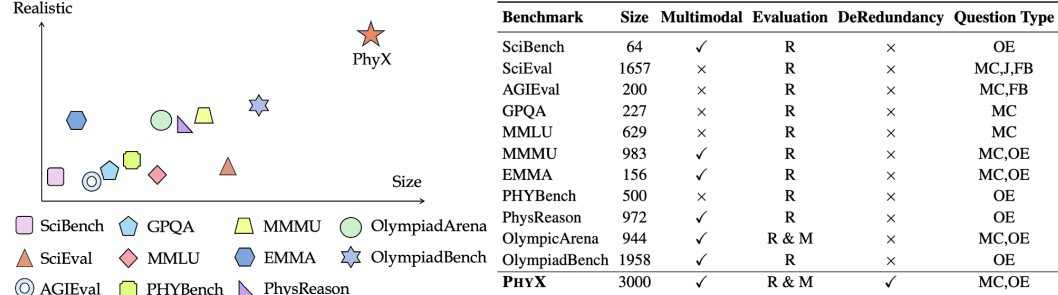

| Benchmark | Size | Multimodal | Evaluation | DeRedundancy | Question Type |
|---|---|---|---|---|---|
| SciBench | 64 | ✓ | R | ✗ | OE |
| SciEval | 1657 | ✗ | R | ✗ | MC,J,FB |
| AGIEval | 200 | ✗ | R | ✗ | MC,FB |
| GPQA | 227 | ✗ | R | ✗ | MC |
| MMLU | 629 | ✗ | R | ✗ | MC |
| MMMU | 983 | ✓ | R | ✗ | MC,OE |
| EMMA | 156 | ✓ | R | ✗ | MC,OE |
| PHYBench | 500 | ✗ | R | ✗ | OE |
| PhysReason | 972 | ✓ | R | ✗ | OE |
| OlympicArena | 944 | ✓ | R & M | ✗ | MC,OE |
| OlympiadBench | 1958 | ✓ | R | ✗ | OE |
| **PHYX** | 3000 | ✓ | R & M | ✓ | MC,OE |

Figure 3: Comparison with existing benchmarks. Realistic refers to the extent to which the dataset contains visually realistic physical scenarios. Size indicates the number of physics questions with images in multimodal benchmarks or total physics questions in text-only benchmarks. For evaluation methods, R: rule-based, M: model-based. For question type, OE: Open-ended, MC: Multiple-choice, FB: Fill-in-the-blank, J: Judgement. In comparison, PHYX leads in all aspects.

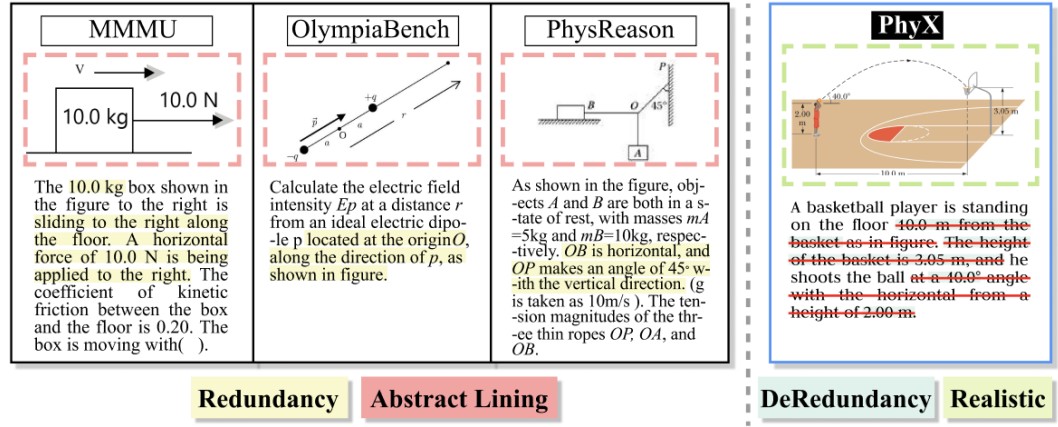

Figure 4: Existing benchmarks that contain physics questions suffer from information redundancy and abstract representation. In contrast, de-redundancy in PHYX increases the difficulty, as models can perceive concepts from ONE modality only. Additionally, realistic visuals challenges models to accurately apply physical laws.

and *hierarchical answer judge* (rule-based and model-based). It also seamlessly integrates with mainstream toolkits (e.g., VLMEvalKit, lmms-eval) for reproducible benchmarking. **Critical Insights on Reasoning:** We provide granular performance analysis and reveal some interesting observations, which sheds light on the design of the future models that jointly consider the disciplinary knowledge, symbolic operations, and real-world constraints for physical reasoning.

## 2 THE PHYX BENCHMARK

### 2.1 OVERVIEW OF PHYX

We introduce PHYX, a novel benchmark meticulously curated to assess the physical reasoning capabilities of foundation models. PHYX consists of 3,000 visually-grounded physics questions, meticulously curated to cover six distinct physics domains including *Mechanics* (550), *Electromagnetism* (550), *Thermodynamics* (500), *Wave/Acoustics* (500), *Optics* (500), and *Modern Physics* (400). Each problem in PHYX is centered around realistic physical scenarios to robustly assess the model's ability to reason the physical world. Detailed data statistics are summarized in Table 1, with representative question examples from each domains illustrated in Figure 2. To enable a comprehensive assessment, each question within PHYX has been categorized into six well-defined physical reasoning types: *Physical Model Grounding Reasoning*, *Spatial Relation Reasoning*, *Multi-Formula Reasoning*, *Implicit Condition Reasoning*, *Numerical Reasoning*, and *Predictive Reasoning*. Detailed definitions and illustrative examples of these reasoning types are provided in Appendix F.4.

Table 1: **Key Statistics of PHYX.**

| Statistic | Number |
|---|---|
| **Total new questions** | **6,000** |
| - Multiple-choice questions | 3,000 (50.0%) |
| - Open-ended questions | 3,000 (50.0%) |
| Unique number of images | 3,000 |
| Unique number of questions | 3,000 |
| Maximum description length | 288 |
| Maximum question length | 119 |
| Maximum option length | 46 |
| Average description length | 48.3 |
| Average question length | 14.6 |
| Average option length | 11.2 |

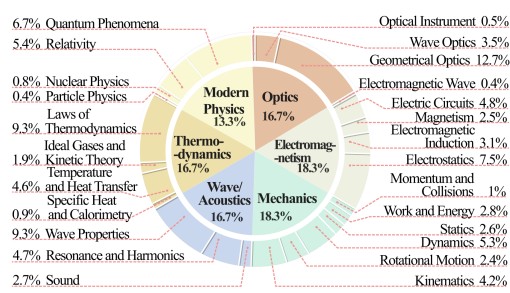

Figure 5: **Fine-grained Distribution of PHYX.**

Through its carefully curated structure and extensive coverage of diverse reasoning dimensions, PHYX represents a robust resource for systematically benchmarking and advancing the capabilities of foundation models in realistic physical reasoning tasks.

## 2.2 DATA CURATION PROCESS

**Data Collection.** To ensure high-quality data, we design a four-stage data collection process. Firstly, we conducted an in-depth survey of core physics disciplines to determine the coverage of our benchmark. We selected diverse physics domains and subfields, and defined a set of reasoning types. Secondly, we recruited a team of graduate students in STEM fields to serve as expert annotators. Annotators are instructed to comply with copyright and licensing rules by avoiding content from sources that restrict copying or redistribution. To mitigate potential data contamination in foundation models, they are advised to select questions for which answers are not immediately available alongside the problem, such as those found in separate materials or at the end of textbooks. Then, each open-ended question is required to be converted into a multiple-choice version, and vice versa. We also constructed three parallel versions of each question: (1) the original version; (2) a concise version where redundant textual information—those duplicated by the corresponding image—was removed; and (3) a question-only version that retains only the core question. Lastly, to support evaluation of LLMs, we used GPT-4o to generate descriptive captions for each image, aim to summarize the visual content in a self-contained textual form. This data curation process results in a diverse collection of 3,300 questions from various sources. The detailed annotation protocol is in Appendix I.

**Data Quality Control.** To further control the quality of our data, we perform a three-stage data cleaning process. First, we detect potentially duplicated questions by analyzing lexical overlap, followed by manual review from physics Ph.D. students to confirm and remove duplicates. Then, we filter out the shortest 10% of questions based on their textual length. This rigorous process plays a crucial role in maintaining the quality and difficulty of PHYX.

## 2.3 KEY DIFFERENCE COMPARED TO EXISTING BENCHMARKS

**Compared with Scientific Knowledge Benchmarks.** From Figure 3, science benchmarks like MMMU (Yue et al., 2024) cover broad disciplinary QA but lack focus on physical reasoning. These benchmarks often rely on basic understanding of disciplinary knowledge, with tasks that prioritize simple factual recall. In contrast, PHYX demands integration of visual cues with implicit physical laws, requiring models to perform context-driven inference. This targeted design evaluates multimodal reasoning about the physical world, exposing gaps in models' ability to handle scientific challenges.

**Compared with Mathematical Reasoning Benchmarks.** Mathematical reasoning benchmarks, such as MathVista (Lu et al.), MathVerse (Zhang et al., 2024), and MATH-V (Wang et al., 2024a), focus on logical deduction with clear expressions and explicit conditions, representing a subset of the challenges in physical reasoning. Physical reasoning extends beyond these by requiring models to model real-world contexts, identify implicit conditions from visual cues (e.g., Figure 6), and integrate the application of physical laws with symbolic logic, which are key capabilities absent in purely mathematical tasks. This makes PHYX a more comprehensive test of multimodal reasoning, capturing the complexity of real-world physics problems.

Table 2: Accuracy scores on the *testmini* subset of PHYX. The highest and the second highest scores of models in each setting are respectively highlighted in blue and red.

| Models | Full-Text | | Text-DeRedundancy | | Text-Minimal | |
|---|---|---|---|---|---|---|
| | *Open-Ended* | *Multi-Choice* | *Open-Ended* | *Multi-Choice* | *Open-Ended* | *Multi-Choice* |
| Random Choice | - | 25 | - | 25 | - | 25 |
| Human Expert (Worst) | - | - | 75.6 | - | - | - |
| Human Expert (Medium) | - | - | 77.8 | - | - | - |
| Human Expert (Best) | - | - | 78.9 | - | - | - |
| *Multimodal Large Language Models* | | | | | | |
| Claude3.7-Sonnet | 44.4 | 65.8 | 42.2 | 64.5 | 17.2 | 41.6 |
| Claude3.5-Sonnet | 40.2 | 62.6 | 39.0 | 63.5 | 17.0 | 43.5 |
| Claude3.5-Haiku | 7.9 | 37.0 | 13.6 | 37.5 | 5.5 | 31.7 |
| GPT-5 | 66.4 | 90.9 | 65.2 | 88.3 | 29.6 | 64.1 |
| GPT-o4-mini | 49.0 | 87.9 | 45.8 | 86.9 | 24.1 | 62.6 |
| GPT-4o | 33.9 | 61.0 | 32.5 | 57.6 | 14.3 | 43.8 |
| Gemini-2.5-Pro | 65.0 | 74.1 | 62.4 | 74.1 | 28.4 | 54.0 |
| InternVL3-78B | 35.9 | 45.6 | 33.1 | 46.9 | 14.8 | 40.5 |
| QVQ-72B-Preview | 17.5 | 40.0 | 17.2 | 40.9 | 7.6 | 33.1 |
| Yi-VL-34B | 3.5 | 34.8 | 3.4 | 34.1 | 1.9 | 34.1 |
| InternVL3-14B | 9.0 | 46.9 | 7.9 | 47.5 | 5.1 | 45.9 |
| InternVL3-8B | 6.3 | 45.5 | 6.5 | 44.9 | 4.6 | 44.0 |
| MiniCPM-o-8B | 7.1 | 31.8 | 7.2 | 31.6 | 3.2 | 34.2 |
| LLaVA-OneVision-7B | 7.2 | 37.7 | 5.7 | 37.3 | 2.7 | 38.0 |
| DeepSeek-VL2-4.5B | 11.4 | 28.2 | 10.2 | 27.8 | 4.7 | 27.3 |
| Kimi-VL-A3B-Instruct-2.8B | 15.6 | 37.1 | 15.4 | 38.7 | 8.1 | 39.3 |
| Kimi-VL-A3B-Thinking-2.8B | 15.3 | 34.4 | 15.8 | 33.2 | 7.4 | 27.0 |
| *Large Language Models* | | | | | | |
| DeepSeek-R1 | 51.8 | 63.1 | 51.2 | 62.9 | 22.2 | 43.6 |
| DeepSeek-V3 | 40.7 | 70.8 | 36.3 | 67.5 | 16.2 | 49.9 |
| GPT-o3-mini | 36.9 | 78.5 | 31.5 | 76.9 | 14.3 | 56.2 |
| Qwen3-4B | 29.6 | 49.2 | 27.5 | 48.4 | 12.1 | 41.8 |
| Qwen3-8B | 32.2 | 50.4 | 31.6 | 48.8 | 13.0 | 37.2 |
| Qwen3-14B | 35.3 | 57.2 | 33.3 | 56.5 | 13.6 | 44.4 |

**Compared with Physics-related Benchmarks** Existing benchmarks (e.g., PHYBench (Qiu et al., 2025), UGPhysics (Xu et al., 2025a), OlympiadBench (He et al., 2024)) prioritize text-based problems or schematic visuals, limiting their assessment of multimodal reasoning. In detail, PHYBench's problems and UGPhysics's questions rely heavily on textual descriptions, while OlympiadBench's problems use simplified diagrams, as shown in Figure 4. These benchmarks mainly test disciplinary knowledge but overlook the integration of visual perception with implicit physical constraints. PHYX bridges these gaps by embedding high-fidelity visual scenarios that require models to decode complex visual cues, infer context-specific physical laws and then reason about problems. Additionally, PHYX mandates equal reliance on both modalities with information de-redundancy, providing a rigorous evaluation of professional-level physical reasoning in MLLMs.

## 3 EXPERIMENTS

### 3.1 EXPERIMENTAL SETUP

**The *testmini* Subset.** PHYX comprises 3,000 high-quality visual physics problems and 18,000 corresponding test instances. To streamline evaluation, we extract a smaller representative subset named *testmini* including 1,000 problems and 6,000 instances. The construction of *testmini* involved a proportional random sampling strategy across different physics domains of PHYX. The quantitative evaluations in all subsequent experiments were conducted on this *testmini* subset, while the full set of results on all 3,000 problems are provided in Appendix E.

**Baselines.** We include random choice as a naive baseline, and we recruit 15 undergraduate and graduate physics students to represent the expert performance baseline, each student was tasked with completing 18 questions. The students were divided into three groups of five, and the results of each group are reported separately. Then, we conduct experiments on (a) Reasoning MLLMs: Gemini-2.5-Pro (Team, 2025), GPT-5 (OpenAI, 2025c), GPT-o4-mini (OpenAI, 2025b), Claude-3.7-Sonnet (claude, 2025), QVQ-Preview (Team, 2024), LLaVA-OneVision (Li et al., 2024), MiniCPM-

Table 3: Average scores by model across different domains of physics with *Open-Ended Text-DeRedundancy* questions. The highest and the second highest scores of models in each domain are respectively highlighted in each setting in red and blue.

| Models | Overall | Mechanics | Electro-magnetism | Thermo-dynamics | Waves & Acoustics | Optics | Modern Physics |
|---|---|---|---|---|---|---|---|
| Human Expert (Worst) | 75.6 | 76.5 | 60.0 | 66.7 | 86.7 | 69.2 | 86.7 |
| Human Expert (Medium) | 77.8 | 94.1 | 53.3 | 60.0 | 93.3 | 76.9 | 86.7 |
| Human Expert (Best) | 78.9 | 76.5 | 86.7 | 73.3 | 86.7 | 69.2 | 86.7 |
| *Multimodal Large Language Models* | | | | | | | |
| Claude3.7-Sonnet | 42.2 | 58.2 | 36.7 | 31.5 | 46.7 | 44.6 | 35.2 |
| Claude3.5-Sonnet | 39.0 | 53.5 | 27.8 | 33.3 | 49.7 | 35.5 | 3.9 |
| Claude3.5-Haiku | 13.6 | 18.8 | 8.9 | 11.5 | 18.8 | 12.0 | 11.5 |
| GPT-5 | 65.2 | 80.2 | 57.4 | 55.2 | 71.0 | 63.3 | 56.5 |
| GPT-o4-mini | 45.8 | 52.3 | 43.2 | 41.8 | 52.7 | 44.0 | 40.6 |
| GPT-4o | 32.5 | 45.9 | 24.3 | 26.1 | 53.9 | 23.5 | 21.2 |
| Gemini-2.5-Pro | 62.4 | 77.6 | 59.2 | 61.8 | 64.8 | 57.2 | 53.3 |
| InternVL3-78B | 33.1 | 48.8 | 27.2 | 25.5 | 43.0 | 28.9 | 24.8 |
| QVQ-72B-Preview | 17.2 | 31.7 | 11.2 | 10.9 | 20.0 | 12.0 | 16.9 |
| Yi-VL-34B | 3.4 | 1.8 | 3.5 | 4.8 | 2.4 | 4.2 | 3.6 |
| InternVL3-14B | 7.9 | 12.4 | 8.9 | 4.2 | 8.5 | 4.8 | 8.5 |
| InternVL3-8B | 6.5 | 10.6 | 6.5 | 3.6 | 4.9 | 6.6 | 6.7 |
| MiniCPM-o-8B | 7.2 | 11.8 | 6.5 | 6.1 | 7.3 | 6.0 | 5.5 |
| LLaVA-OneVision-7B | 5.7 | 10.6 | 4.1 | 6.1 | 7.3 | 3.0 | 3.0 |
| DeepSeek-VL2-4.5B | 10.2 | 16.5 | 7.1 | 10.3 | 13.3 | 9.0 | 4.8 |
| Kimi-VL-A3B-Instruct-2.8B | 15.4 | 20.6 | 10.1 | 13.3 | 20.0 | 16.2 | 12.1 |
| Kimi-VL-A3B-Thinking-2.8B | 15.8 | 25.9 | 15.4 | 7.9 | 20.6 | 13.3 | 11.5 |
| *Large Language Models* | | | | | | | |
| DeepSeek-R1 | 51.2 | 71.8 | 53.2 | 41.8 | 53.9 | 39.8 | 46.1 |
| DeepSeek-V3 | 36.3 | 52.9 | 39.6 | 28.5 | 36.4 | 28.9 | 30.9 |
| GPT-o3-mini | 31.5 | 41.8 | 24.9 | 23.6 | 32.1 | 33.7 | 32.7 |
| Qwen3-4B | 27.5 | 42.9 | 23.7 | 21.2 | 35.8 | 21.1 | 20.0 |
| Qwen3-8B | 31.6 | 51.2 | 26.6 | 19.4 | 37.0 | 29.5 | 25.5 |
| Qwen3-14B | 33.3 | 52.9 | 30.8 | 18.2 | 40.0 | 27.1 | 30.3 |

o (Yao et al., 2024), Kimi-VL-A3B-Thinking (Team et al., 2025), (b) General MLLMs: GPT-4o (OpenAI, 2024a), Claude-3.5-Sonnet (claude, 2024b), Claude-3.5-Haiku (claude, 2024a), InternVL3 (Zhu et al., 2025), Yi-VL (Young et al., 2024), Kimi-VL-A3B-Instruct (Team et al., 2025), (c) LLMs: GPT-o3-mini (OpenAI, 2025a), DeepSeek-R1 (Guo et al., 2025), DeepSeek-V3 (DeepSeek-AI, 2025), Qwen3 (Yang et al., 2025), augmented with image captions generated by GPT-4o.

## 3.2 EVALUATION PROTOCOLS

Our evaluation is conducted with Chain-of-Thought (CoT) prompting to assess the reasoning capability of models. For both open-ended (OE) and multiple-choice (MC) questions, the instruction-following capabilities of models can vary significantly. To this end, we design a universal evaluation pipeline for all recent LLMs and MLLMs with different instruction-following capabilities:

**Step 1. Prediction Generation.** Initially, the models generate predictions given the input query, which incorporates different problem descriptions according to the specific settings, the question, and the image, using the template defined in Appendix G.1.

**Step 2. Answer Extraction.** The raw predictions often contain reasoning steps, explanations, or irrelevant conversational filler. To precisely extract the definitive answer from these raw outputs, we separately employ rule-based answer extraction strategies, which are detailed in Appendix G.2.

**Step 3. Answer Judgment.** For OE questions, the next step is comparing the extracted answer against the ground truth. Given that answers in OE physics questions can be expressed in myriad ways, we propose an evaluation mechanism using an LLM, such as DeepSeek-V3 (DeepSeek-AI, 2025), as a judge, using the template defined in Appendix G.3. We feeds the answer extracted and the ground truth to an LLM multiple times and checks if an LLM succeed in all attempts. A preliminary study of 200 examples shows that DeepSeek-V3 can judge the answer with more than 99% accuracy with an affordable cost. For MC questions, we first attempt to directly match the option letter. If it fails, we then use an LLM as a judge, using the template for OE questions.

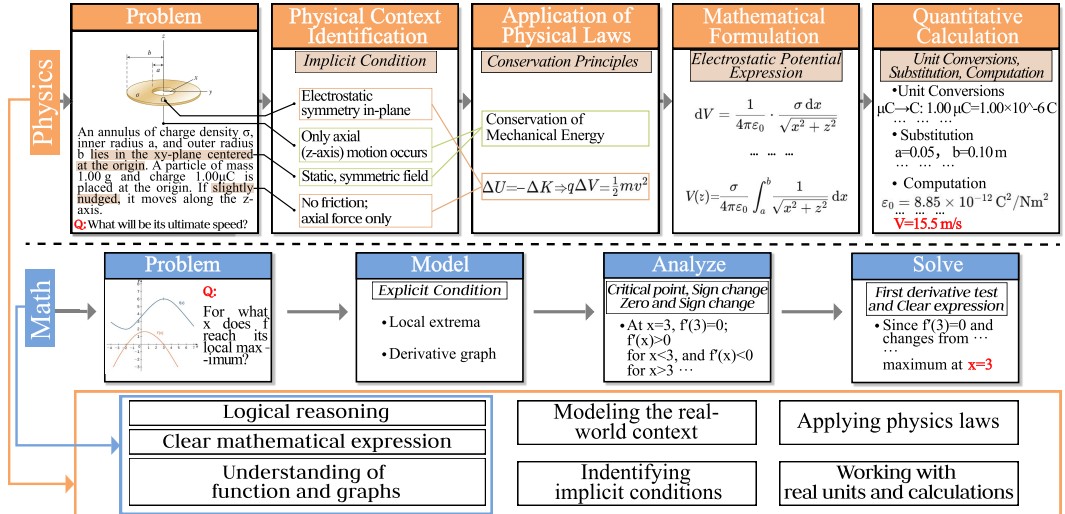

Figure 6: An real example of reasoning trajectory based on GPT-4o and the comparison of required capabilities when solving physical and mathematical problems.

## 3.3 MAIN RESULTS

In this section, we present a comprehensive comparison of LLMs and MLLMs on the PHYX benchmark, detailed in Table 2 and Table 3. Our key findings can be summarized as follows:

**Challenging Nature of PHYX.** PHYX presents significant challenges for current models. Notably, even the worst human experts achieve accuracy of 75.6%, significantly outperforming all the models included in our comparative analysis. This disparity demonstrates an existing gap between human expertise and current model capabilities, reflecting the demanding standards inherent in PHYX.

**Question Format as a Determinant of Model Discriminability.** The result reveals that multiple-choice questions reduce the performance gap across models, enabling weaker models to rely on surface-level cues. In contrast, open-ended questions demand genuine reasoning and precise answer generation, leading to greater differentiation between models. This suggests that the open-ended format provides higher discriminative power when evaluating multimodal reasoning capabilities.

**Domain-specific Variability in Multimodal Reasoning.** As shown in Table 3, in domains such as Waves/Acoustics and Mechanics, which typically include natural images and questions requiring relatively less reasoning, models tend to achieve higher performance. Conversely, in domains such as Thermodynamics and Modern Physics, where tasks frequently demand intricate visual perception and multi-step reasoning, models performance is generally lower.

**Reasoning-oriented Models Perform Better.** Leading reasoning-oriented models such as GPT-o4-mini and DeepSeek-R1 achieve accuracies of 45.8% and 51.2%, respectively, significantly outperforming general-purpose models like GPT-4o and Claude3.7-Sonnet. The results highlight the advantage of models specifically optimized for reasoning tasks, suggesting that architectural and training differences play a key role in bridging the multimodal reasoning gap.

## 3.4 DISCUSSION

**Mathematical Reasoning Is Not Enough.** Comparing GPT-4o's performance on PHYX to its previously reported results on MathVista (63.8%) and MATH-V (63.8%), we observe substantially lower accuracy in physical reasoning tasks, underscoring that these tasks present challenges that go beyond mathematical abstraction. As shown in Figure 6, unlike mathematics problems, where symbolic manipulation and abstraction are often sufficient, physical reasoning requires models to decode implicit conditions in the problem statement (e.g., interpreting "smooth surface" as implying zero friction), ground physical laws in concrete visual and material contexts (e.g., recognizing whether a wooden block will float or whether a surface is rough or smooth), and maintain internal

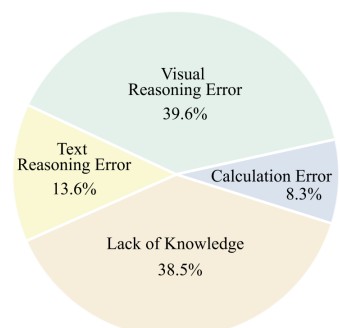
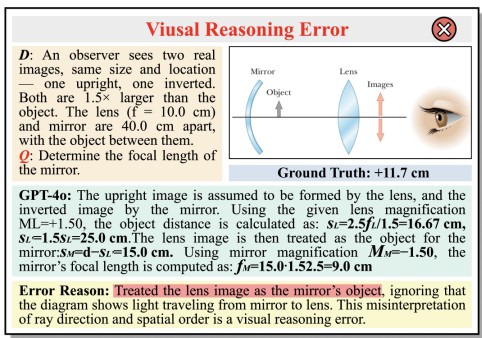

Figure 7: The error distribution over 90 annotated errors with a typical visual reasoning error, which is easy for humans but challenging for GPT-4o. More examples can be found in the Appendix H.

consistency across reasoning chains, since the laws of physics remain invariant regardless of the reasoning trajectory. This tight integration of perception, abstraction, and dynamic simulation makes physical reasoning qualitatively more demanding, revealing a fundamental barrier that current LLMs and MLLMs have yet to overcome.

**Impact of Redundancy Removal on Models.** We observe that most MLLMs exhibit performance degradation under the Text-DeRedundancy setting, reflecting that most MLLMs are built upon a powerful language model backbone, and thus tend to over-rely on textual information while under-utilizing visual input. However, certain models (e.g., Claude 3.5-Haiku) benefit from redundancy removal, which aligns with recent findings (Li et al., 2025; Mao et al., 2025) that imperfect cross-modal alignment can introduce noisy or conflicting signals. In such cases, redundant textual cues exacerbate alignment issues, and their removal clarifies multimodal inputs. Notably, this effect is unique to MLLMs and does not occur in pure-text LLMs, further suggesting that the root cause lies in cross-modal fusion rather than general reasoning ability.

**MLLMs' Physical Reasoning Relies More on Text.** Our experiments show a clear performance gradient across the three input variations: Full-Text, Text-DeRedundancy, and Text-Minimal, with decreasing accuracy in that order. This indicates that most MLLMs rely heavily on detailed textual descriptions, highlighting their limited ability to reason purely from visual context.

**Competitive LLMs Highlight Limitations in Multimodal Fusion.** Despite lacking direct visual input, LLMs such as DeepSeek-R1 and GPT-o3-mini perform competitively with most multimodal models. The strong performance of LLMs suggests that, in many cases, the caption provides sufficient visual context for reasoning. This highlights both the impressive generalization capabilities of LLMs and the current limitations of MLLMs in leveraging raw visual signals for physical reasoning.

### 3.5 ERROR ANALYSIS

To dive into the reasoning capabilities and limitations of models, we meticulously inspected 96 randomly sampled incorrect predictions and performed an in-depth analysis based on GPT-4o. The objectives of this analysis were twofold: to identify current model weaknesses and to guide future enhancements in model design and training. The distribution of these errors is illustrated in Figure 7, and a comprehensive case study of 30 notable cases is included in Appendix H.

**Reasoning Errors (53.2%)** encompass both visual and textual reasoning failures. While *visual reasoning errors* typically arise from incorrect extraction, spatial relationships, or reasoning based on visual information (e.g., misreading the voltage in Appendix 8), *textual reasoning errors* are characterized by misinterpretation of textual content, such as overlooking explicit conditions (e.g., ignoring the instruction to neglect friction in Appendix 4). Furthermore, our analysis reveals three deeper challenges in multimodal reasoning:

- *Context Switching.* Rapidly transitioning between textual and visual modalities can cause models to lose focus or misinterpret key data. Prior work (Zhang et al., 2025; Li et al., 2025) confirms that modality switching introduces significant cognitive load, leading to unstable attention, misalignment in cross-modal representations, and disrupted reasoning chains, especially in tasks requiring deep integration of visual and textual information.

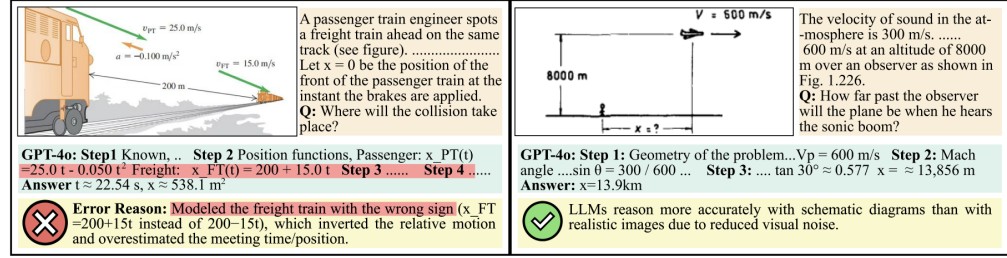

Figure 8: Both are relative motion problems, easy for human experts. ***Left***: Error case from PHYX with realistic images, which introduce visual noise and make reasoning harder. ***Right***: Correct case from concurrent work SeePhys (Xiang et al., 2025), where schematics image simplify perception.

- ***Cross-Modal Reasoning Difficulty.*** Models often struggle to deeply integrate visual and textual modalities, particularly when fine-grained visual cues must align with detailed textual descriptions. Existing studies (Mao et al., 2025; Yue et al., 2024) highlight that current fusion strategies frequently yield superficial alignment, limiting cross-modal reasoning.

- ***Visual Realism Challenge.*** As illustrated in Figure 8, unlike other benchmarks that primarily employ abstract line drawings or schematic sketches, our dataset contains realistic images. This realism significantly increases perceptual and reasoning difficulty: models must construct an abstract physical problem from a noisy, unstructured visual input rather than relying on simplified shapes.

**Lack of Knowledge (38.5%)** reflects GPT-4o's incomplete understanding of physical domain knowledge. As shown in Appendix 25, the model lacks the fundamental knowledge. Specifically, it ignores that the slower speed in the liver requires a correction when estimating depth from the reflection geometry, leading to an overestimated result.

**Calculation Error (8.3%)** refer to mistakes in arithmetic operations, formula application, or unit conversions. These errors indicate that the model has grasped the physical context and relevant concepts but fails in the final step of numerical computation.

## 4 RELATED WORK

**Multi-modal Large Language Models.** Multi-modal large language models (MLLMs) (OpenAI, 2025b; Team, 2025) have shown great potential and achieved excellent visual understanding by integrating visual and textual data across a wide range of multimodal tasks. Recent advances in LLMs have motivated efforts (Wei et al., 2022; Ouyang et al., 2022) to explore MLLM reasoning. Despite such achievements, it remains unclear whether these models truly possess advanced reasoning abilities, especially in the physical area that is closer to the real world. To bridge this gap and evaluate the physical reasoning capabilities of MLLMs, we introduce PHYX, a multimodal benchmark to evaluate the real reasoning ability of recent advanced MLLMs in physics.

**MLLM Benchmarks.** Recently, several MLLM scientific benchmark (Yue et al., 2024; Wang et al., 2024b; He et al., 2024; Huang et al., 2024; Zhang et al., 2025; Hao et al., 2025) have also been proposed. For example, PhysReason (Zhang et al., 2025) includes a multimodal subset of 972 physics problems with figures to evaluate the MLLMs. EMMA (Hao et al., 2025) comprises 2,788 problems covering various scientific areas such as mathematics, physics, and coding. However, all of these benchmarks contain only a small subset of data in physics, which still cannot fully evaluate MLLM's ability on physical reasoning. More related works are discussed in Appendix D.

## 5 CONCLUSION

Existing benchmarks have overlooked the critical task of physical reasoning, which requires integrating domain knowledge, symbolic reasoning, and real-world constraints. To address this, we present PHYX, the first large-scale benchmark for evaluating physical reasoning in multimodal, visually grounded scenarios. Through rigorous evaluation, we reveal that state-of-the-art models exhibit significant limitations in physical reasoning. Our findings highlight the urgent need for future models to improve deep physical reasoning over surface-level associations.

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

# Table of Contents in Appendix

## A  THE USE OF LARGE LANGUAGE MODELS

In preparing this manuscript, we used a Large Language Model (LLM) solely to assist with minor language polishing and improvements in readability. The LLM did not contribute to research ideation, analysis, or substantive writing. All scientific content and conclusions are entirely the responsibility of the authors.

## B  REPRODUCIBILITY STATEMENT

We have made every effort to ensure the reproducibility of our results. All code, scripts, and dataset used in this work are available in an anonymous repository anonymous.4open.science. Detailed descriptions of the benchmark setup, data collection, and preprocessing procedures are provided in Section 3 of the main text and in Appendix G. With these resources, all reported results can be fully reproduced.

## C  ETHICS STATEMENT

**Legal Compliance.** All questions included in PHYX are sourced from publicly accessible materials. During data collection, annotators are instructed to strictly follow the copyright and licensing terms of the original platforms. Any content from sources that prohibit reuse or redistribution MUST be explicitly excluded. PHYX is a non-commercial project, and its usage aligns with the principles outlined in Fair Use §107: "the fair use of a copyrighted work, including such use by ...... scholarship, or research, is not an infringement of copyright", where fair use is determined by "the purpose and character of the use, including whether such use is of a commercial nature or is for nonprofit educational purposes" and "the effect of the use upon the potential market for or value of the copyrighted work."

**Dataset Intended Usage and License.** The full details of the PHYX dataset are presented in this paper, and both the PHYX and code for reproducing results will be made publicly available. The PHYX dataset is not supposed to be used to train models for cheating. The primary goal is to support the research community in benchmarking and advancing physical reasoning in LLMs and MLLMs. We take full responsibility for any rights violation that may arise. Both the PHYX data and our open-source code are released under the MIT license.

## D  MORE RELATED WORK

Several LLM benchmarks (Hendrycks et al.; Sun et al., 2024; Rein et al., 2024; Austin et al., 2021; Zhou et al., 2023) have been proposed to evaluate LLM's ability on various aspects. Among these works, the most related one is PHYBench (Qiu et al., 2025), which also focuses in the physic reasoning area. Although evaluating the same discipline, their scope remains narrow since it includes only a small number of questions, making it insufficient to fully assess a model's reasoning capabilities. Furthermore, PHYBench concentrates exclusively on evaluating through text. However, in real-world scenarios, solving physics problems also requires visual perception and interpretation. Concurrently, three related efforts (Wang et al., 2025; Xiang et al., 2025; Xu et al., 2025b) have emerged. While sharing a similar motivation, these benchmarks rely on schematic images that simplify visual perception. In contrast, our dataset leverages realistic images, which introduce visual noise and thereby make the reasoning process more challenging and closer to real-world conditions.

## E  MORE EXPERIMENTAL RESULTS

Table 4 reports the accuracy scores of the leading MLLM (GPT-4o) on the full *test* subset. The minor differences between overall scores on the *test* subset and the *testmini* subset, suggest that the *testmini* subset effectively mirrors the *test* subset, serving as a valuable evaluation subset for model development, especially for those who have limited computing resources.

Table 4: Accuracy scores by GPT-4o across different domains of physics with open-ended text de-redundancy questions on the *test* subset and *testmini* subset of PHYX.

| GPT-4o | Overall | Mechanics | Electro-magnetism | Thermo-dynamics | Waves & Acoustics | Optics | Modern Physics |
|---|---|---|---|---|---|---|---|
| *testmini* | 32.5 | 45.9 | 24.3 | 26.1 | 53.9 | 23.5 | 21.2 |
| *test* | 34.0 | 45.8 | 37.1 | 20.0 | 38.2 | 27.2 | 34.3 |

# F MORE DATASET DETAILS

## F.1 QUESTION DISTRIBUTION

All questions in PHYX are written in English. Figure 9 presents the distribution of word counts of questions in the Text-DeRedundancy setting, demonstrating the variation in question lengths. The similarity between the median and average word counts suggests a roughly symmetrical distribution.

## F.2 INTRODUCTION OF DOMAIN AND SUBFIELD

As shown in Table 5, PHYX covers 6 core domains and 25 subdomains.

**Mechanics.** Mechanics is the branch of physics concerned with the motion of objects and the forces that cause or change this motion. It encompasses both classical mechanics and key subfields such as *Kinematics* (e.g., velocity, acceleration, free fall), *Dynamics* (e.g., Newton's laws, force analysis, friction), *Work and Energy* (e.g., work-energy theorem, mechanical energy conservation), *Momentum and Collisions* (e.g., conservation of momentum, elastic and inelastic collisions), *Rotational Motion* (e.g., torque, angular acceleration, moment of inertia), and *Statics* (e.g., torque balance, structural analysis). Mechanics lays the groundwork for much of physics, enabling the understanding of how and why objects move or remain at rest in various physical systems.

**Electromagnetism.** Electromagnetism explores the interactions between electric charges and magnetic fields. It includes the subfields of *Electrostatics* (e.g., Coulomb's law, electric fields and potential), *Electric Circuits* (e.g., Ohm's law, circuit analysis, RC circuits), *Magnetism* (e.g., magnetic fields, Lorentz force, Ampère's law), *Electromagnetic Induction* (e.g., Faraday's law, Lenz's law, inductance), and optionally *Maxwell's Equations and Electromagnetic Waves* for advanced topics. This domain underpins much of modern technology, including electric circuits, motors, and wireless transmission.

**Thermodynamics.** Thermodynamics is the study of heat, energy, and their transformations. Its subtopics include *Temperature and Heat Transfer* (e.g., conduction, convection, radiation), *Specific Heat and Calorimetry* (e.g., phase changes, heat calculations), *Laws of Thermodynamics* (e.g., energy conservation, entropy), and *Ideal Gases and Kinetic Theory* (e.g., gas laws, internal energy, pressure). This domain is central to engines, thermal systems, and understanding natural processes.

**Wave/Acoustics.** This domain investigates wave behavior and sound phenomena. Core subfields include *Wave Properties* (e.g., speed, frequency, wavelength, interference), *Sound* (e.g., pitch, loudness, Doppler effect, standing waves), and *Resonance and Harmonics* (e.g., resonant frequencies, vibrations in strings and air columns). These concepts are crucial in fields ranging from acoustics to telecommunications.

**Optics.** Optics studies the behavior and properties of light. It includes *Geometrical Optics* (e.g., reflection, refraction, lens imaging, total internal reflection), *Wave Optics* (e.g., interference, diffraction, polarization), and *Optical Instruments* (e.g., microscopes, telescopes, image formation). Optics has broad applications in imaging, vision science, and photonics.

**Modern Physics.** Modern Physics addresses phenomena beyond the scope of classical mechanics. Its key subfields include *Relativity* (e.g., time dilation, mass-energy equivalence), *Quantum Phenomena* (e.g., photoelectric effect, atomic models), *Nuclear Physics* (e.g., radioactivity, nuclear reactions, mass defect), and optionally *Particle Physics* (e.g., elementary particles, the Standard Model). These topics form the theoretical basis of contemporary physics and technology.

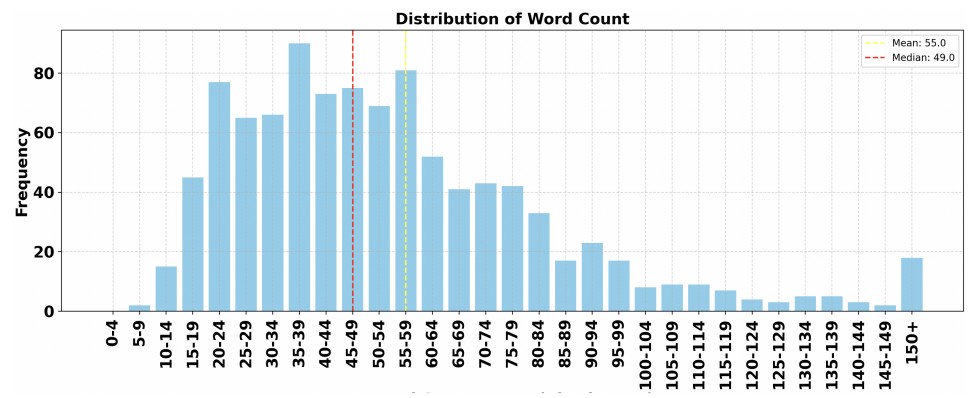

Figure 9: The distribution of the number of words per question in PHYX.

| Domain | Subfields |
|---|---|
| Optics | Optical Instrument, Wave Optics, and Geometrical Optics |
| Electromagnetism | Electromagnetic Wave, Electric Circuits, Magnetism, Electromagnetic Induction, and Electrostatics |
| Mechanics | Momentum and Collisions, Work and Energy, Statics, Dynamics, Relational Motion, and Kinematics. |
| Wave/Acoustics | Sound, Resonance and Harmonics, and Wave Properties |
| Thermodynamics | Specific Heat and Calorimetry, Temperature and Heat Transfer, Ideal Gases and Kinetic Theory, and Laws of Thermodynamics |
| Modern Physics | Particle Physics, Nuclear Physics, Relativity, and Quantum Phenomena |

Table 5: Subfields included in each domain in PHYX.

### F.3 IMAGES BY DOMAINS

In this section, we present example images from the physics problems in PHYX. Figure 10, Figure 11, Figure 12, Figure 13, Figure 14 and Figure 15 show images from problems under the categories of Mechanics, Electromagnetism, Thermodynamics, Wave/Acoustics, Optics, and Modern Physics, respectively.

We observe that the images in our dataset are highly realistic, often depicting concrete physical scenarios rather than stylized or abstract illustrations. While they are not real-world photographs, these visuals are grounded in plausible physical settings. This realism provides essential context for physical reasoning and helps bridge the gap between abstract physics principles and their real-world manifestations.

Across domains, the visual characteristics vary in alignment with the nature of the physical concepts. Despite their domain-specific variations, a unifying theme across all categories is the consistent use of realistic and context-rich imagery, which provides essential grounding for physical interpretation and distinguishes our benchmark from other datasets with overly synthetic or schematic visual content.

### F.4 PHYSICAL REASONING DEFINITION

Six physical reasoning types are defined in Table 6.

## G MORE EVALUATION DETAILS

We conduct all experiments on NVIDIA A100 80G GPUs.

# Mechanics

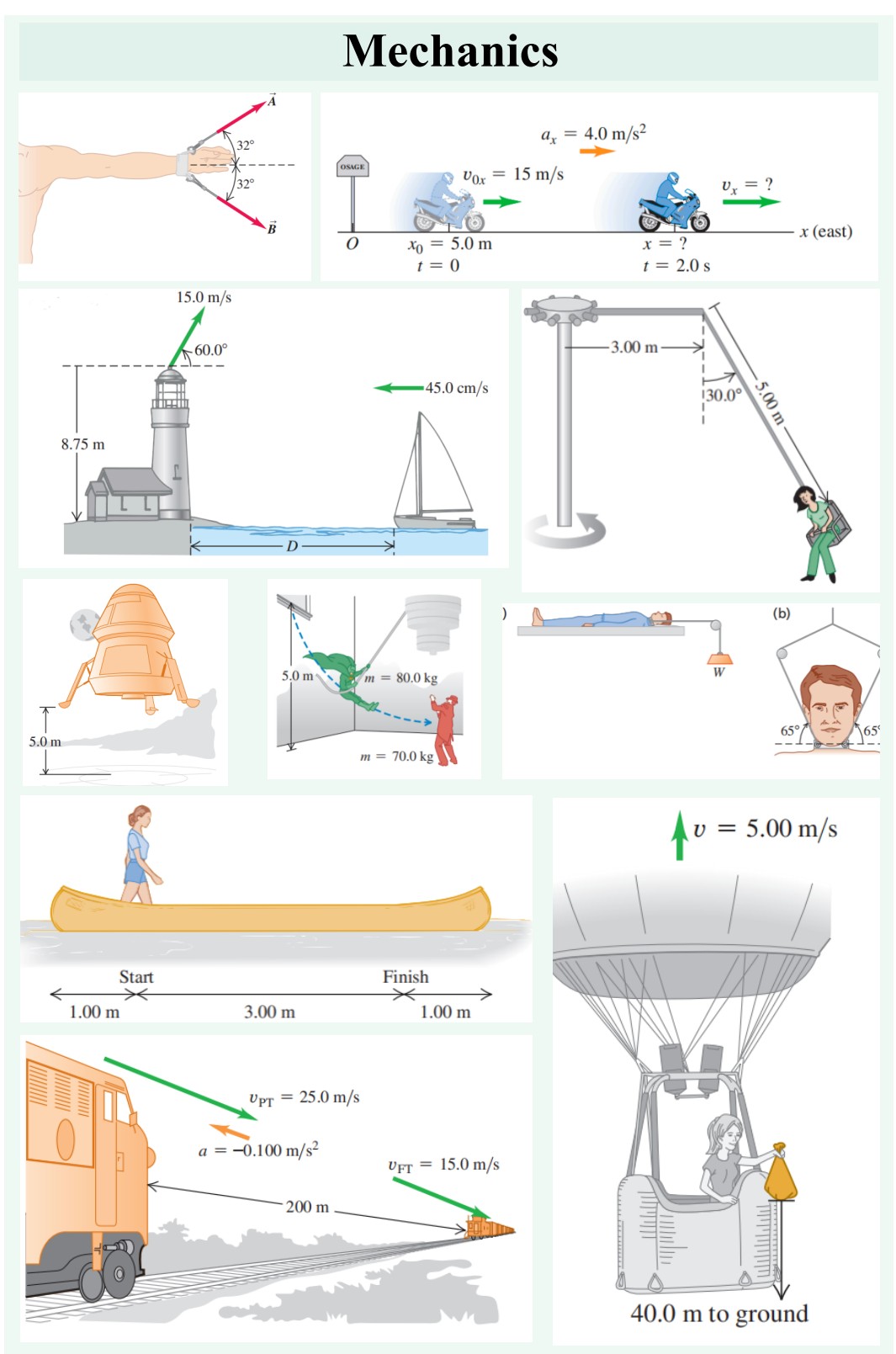

Figure 10: Examples of the visual context for the *Mechanics* domain.

# Electromagnetism

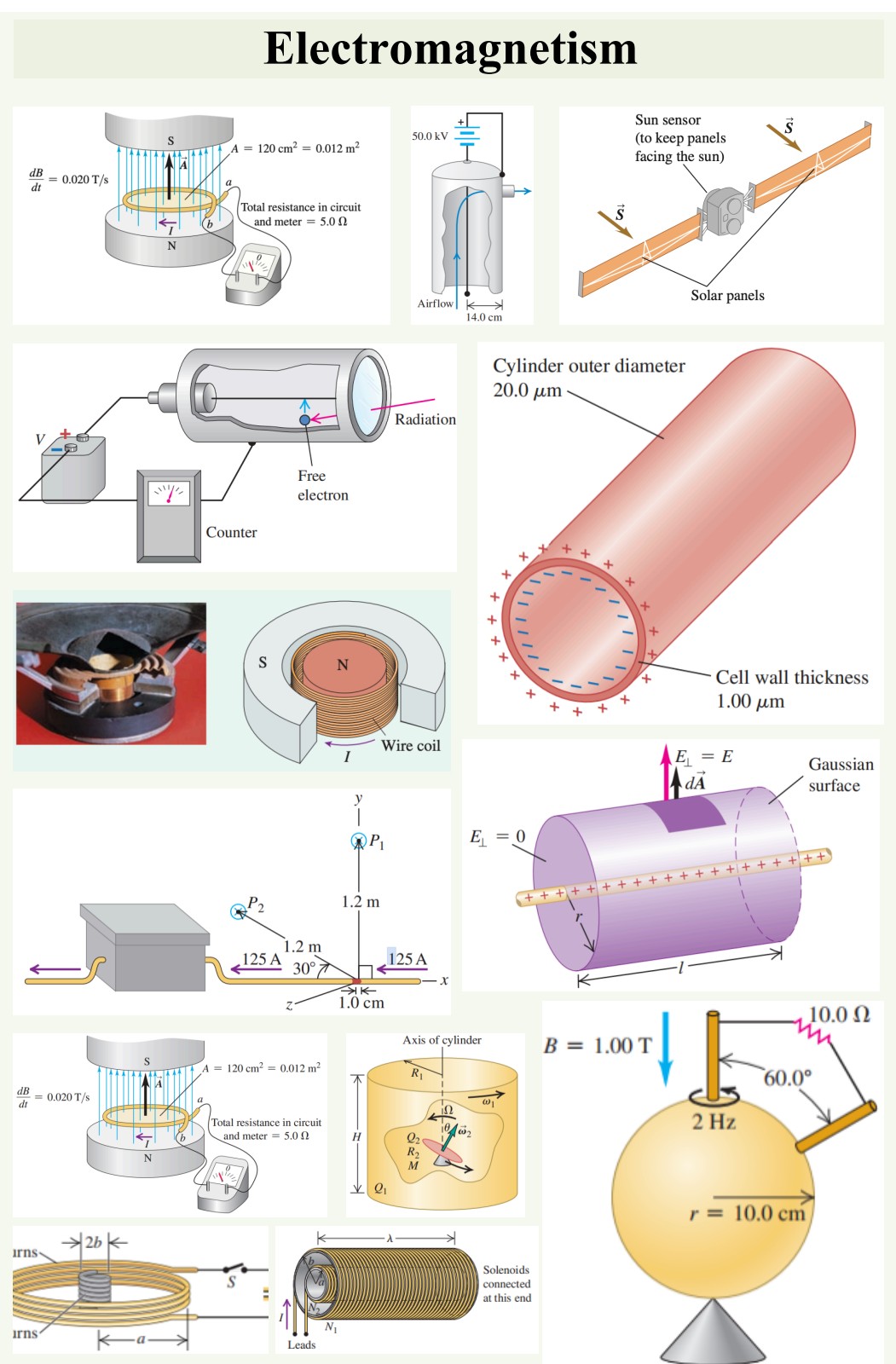

Figure 11: Examples of the visual context for the *Electromagnetism* domain.

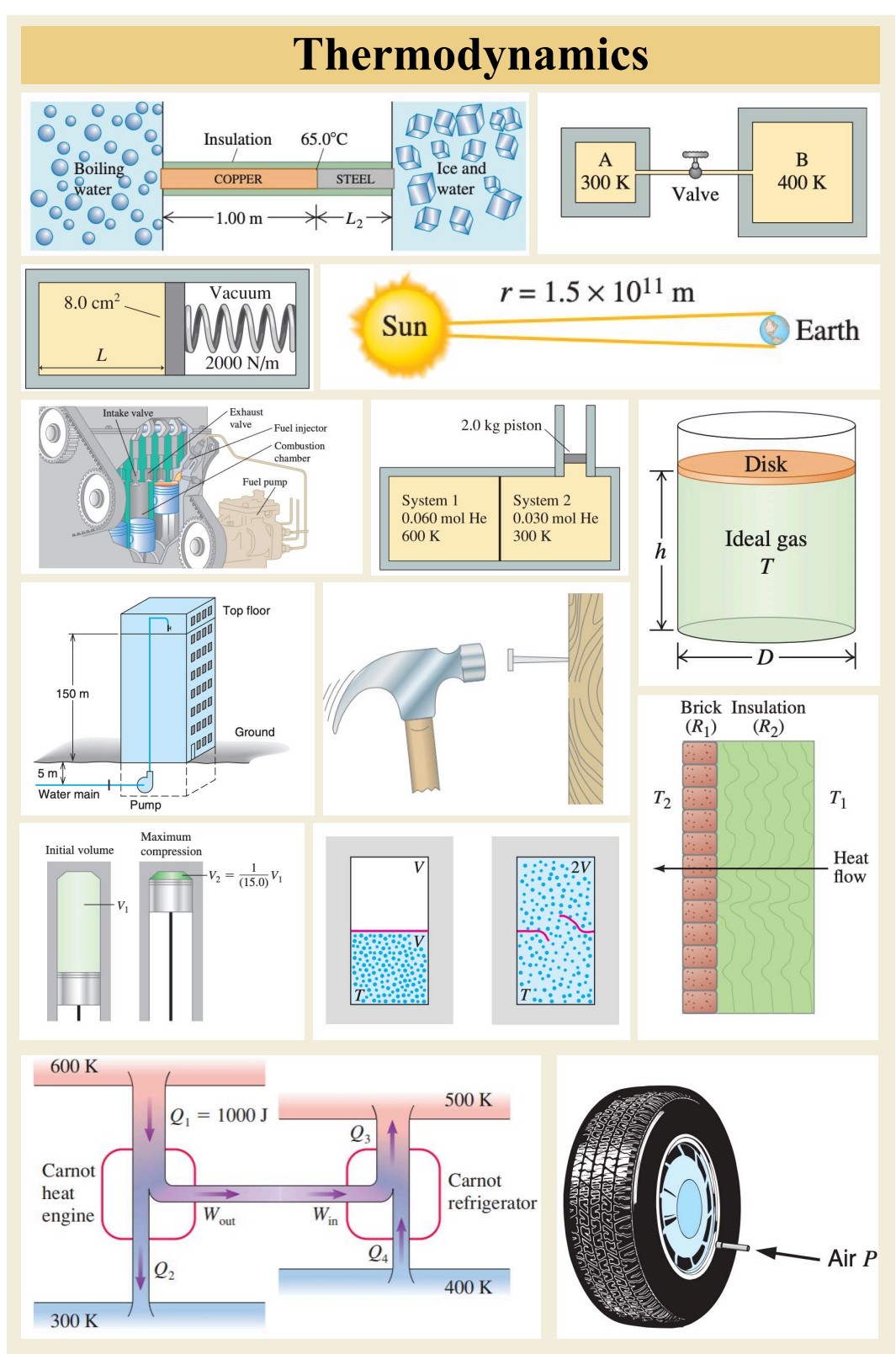

Figure 12: Examples of the visual context for the *Thermodynamics* domain.

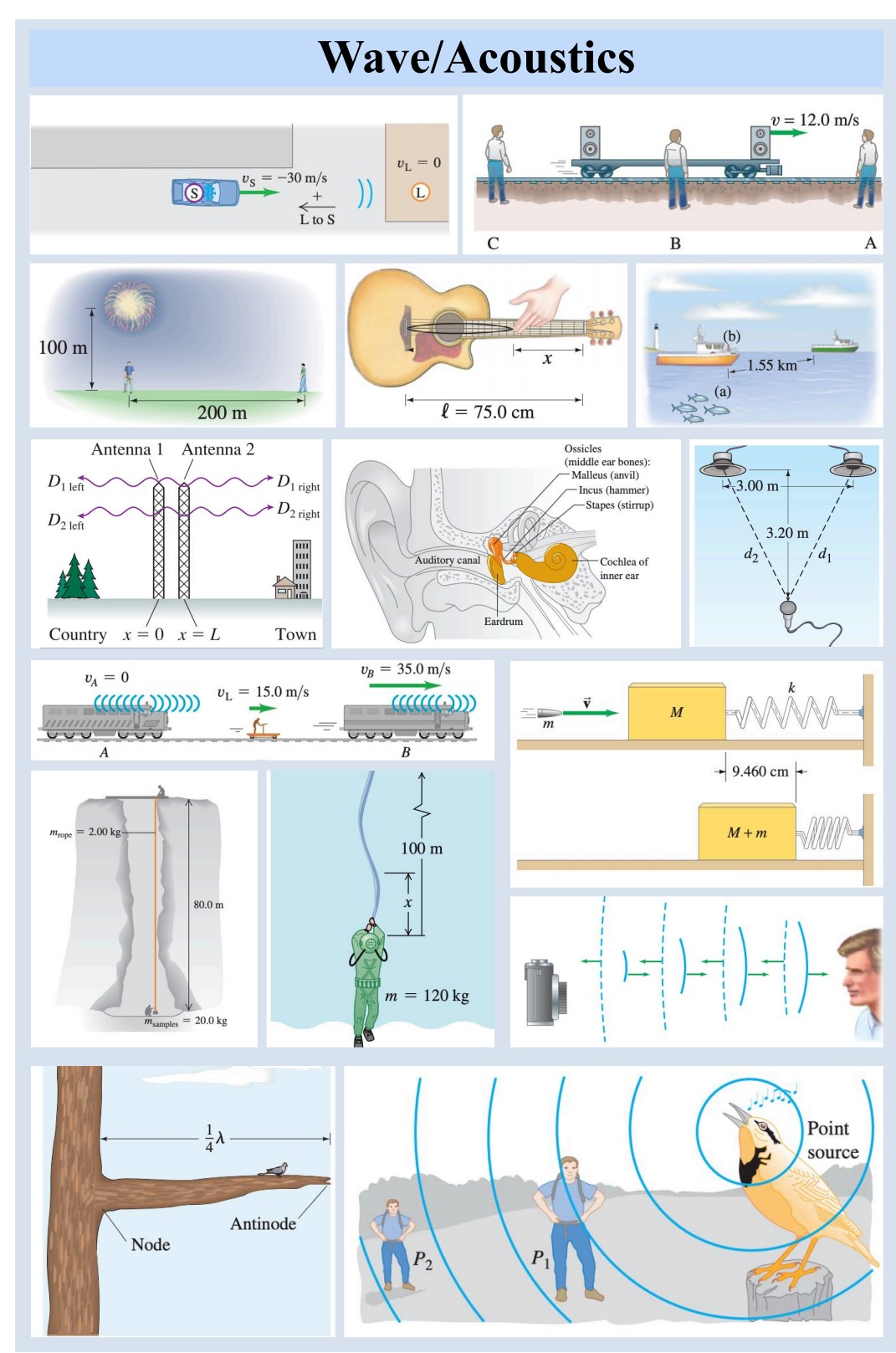

Figure 13: Examples of the visual context for the *Wave/Acoustics* domain.

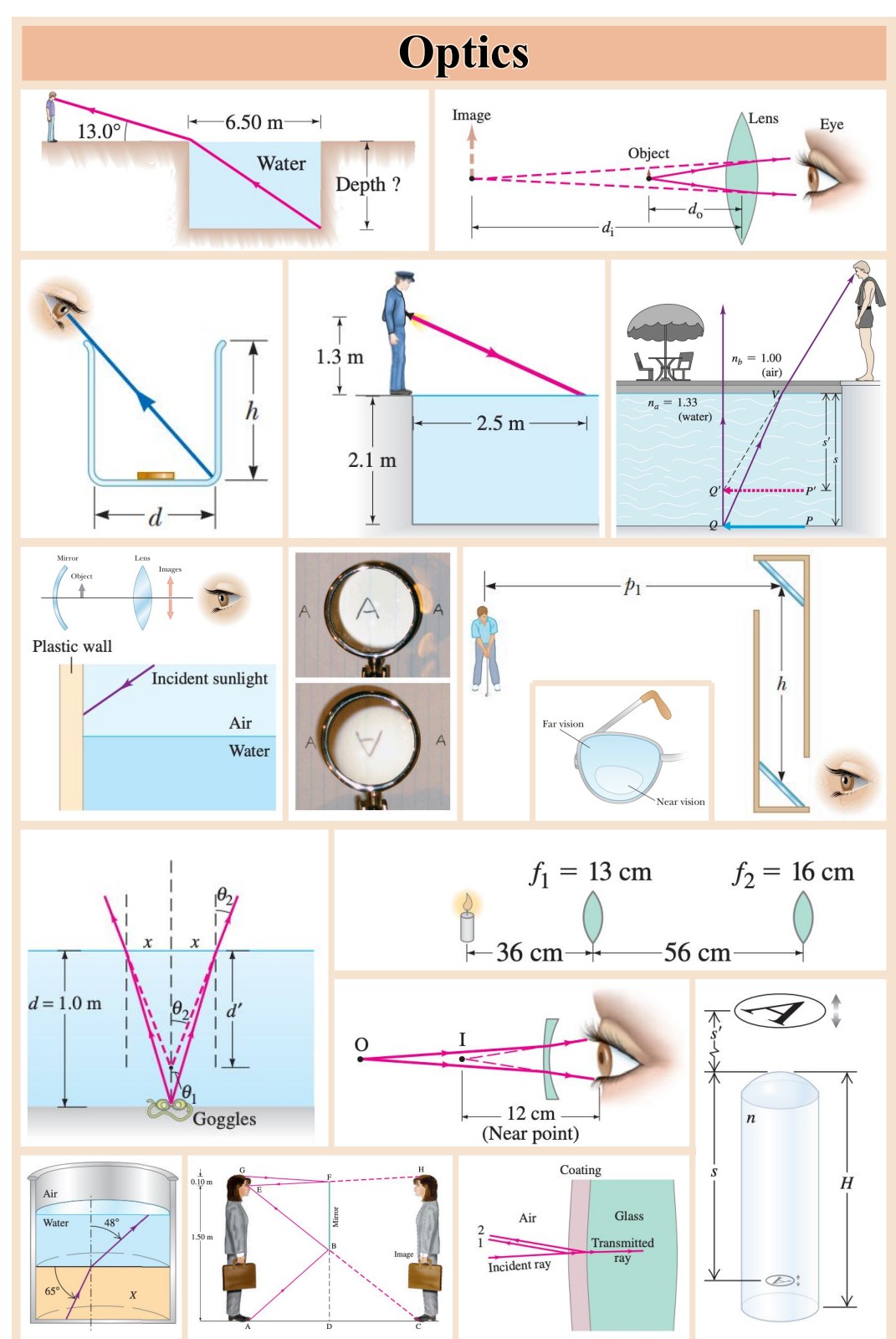

Figure 14: Examples of the visual context for the *Optics* domain.

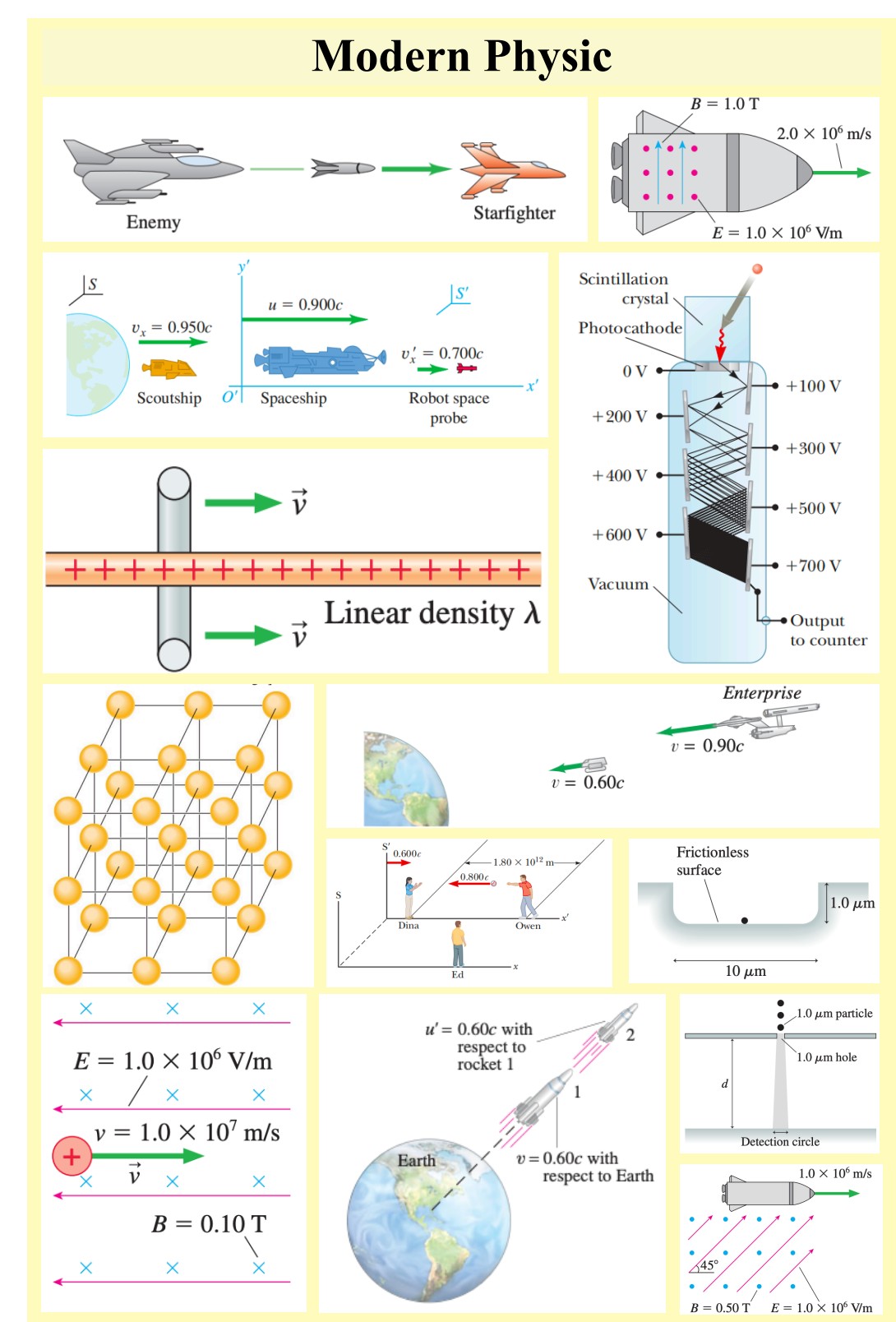

Figure 15: Examples of the visual context for the *Modern Physics* domain.

## CoT Prompting for Generating Answer

```
Please answer the question with step by step reasoning.
```

Figure 16: CoT prompting for generating answer.

## Rule-based Answer Extraction (MC)

```
def MetaPhyX_process_line_MC(line):
    ret = {}

    answers = str(line['answer'])

    ret["index"] = line["index"]
    ret['gt'] = answers
    ret['pred'] = line['prediction'].strip()

    pattern = r'\b(?:correct|answer|option|Answer|Option|
Correct)\b[\s\S]*?([A-D])'
    match = re.search(pattern, ret['pred'])
```

Figure 17: Rule-based answer extraction strategy for MC questions.

### G.1 CoT PROMPTING FOR GENERATING ANSWER

The CoT prompting for generating answer is shown in Figure 16.

### G.2 RULE-BASED ANSWER EXTRACTION

The rule-based answer extraction strategies for MC and OE questions are shown in Figure 17 and Figure 18, respectively.

### G.3 PROMPT FOR ANSWER JUDGE

The prompt for answer judge is shown in Figure 19.

### G.4 PROMPT FOR CAPTION GENERATION

The prompt for caption generation is shown in Figure 20

### G.5 PROMPT FOR REASONING TYPE LABELING

The prompt for reasoning type labeling is shown in Figure 21 and Figure 22

## Rule-based Answer Extraction (OE)

```python
def MetaPhyX_process_line(line):
    ret = {}

    answers = str(line['answer'])

    ret["index"] = line["index"]
    ret['gt'] = answers
    ret['pred'] = line['prediction'].strip()

    pattern = r'\b(?:final\s+answer|
correct\s+answer)\b[^:：]*[:：]\s*(.*?)(?=\n\n\n|\Z)'
    flags = re.IGNORECASE | re.DOTALL
    match = re.search(pattern, ret['pred'], flags=flags)
```

Figure 18: Rule-based answer extraction strategy for OE questions.

## Prompt for Answer Judge

```
Please read the following example.
Given predicted answer and ground truth answer, compare the these two
answers, then ONLY output judegement 1/0 for matched/unmatched at the
end of the prompt.
If the meaning is expressed in the same way, it is also considered
consistent, for example, 0.5m and 50cm.
If the given predicted mentions "approximately", then allow the
Approximation Error, such as 0.49 and approximately 0.5, 0.81 and
approximately 0.8.
Ground truth answer: 26.7kg \n
Predicted answer: The mass of block \( B \) is:
\[
\boxed{26.7 \, \text\{kg\}}
\] \n
Judegement: 1
Ground truth answer: 46.3 kN \n
Predicted answer: The tension \( T_B \) in the cable is approximately:
\[
\boxed{46300 \, \text{N}}
\] \n
Judegement: 1
Ground truth answer: 12 m/s \n
Predicted answer: The speed of the box after 2.00 seconds is:
\[
\boxed{11.3 \, \text{m/s}}
\] \n
Judegement: 0
Ground truth answer: 36.00 kg \n
Predicted answer: The mass of the hanging block \( m_2 \) must be
approximately:
\[
\boxed{36.1 \, \text\{kg\}}
\] \n
Judegement: 1
Ground truth answer: 4.7 m \n
Predicted answer: The stuntman and villain slide approximately **4.69
meters**.
Judegement: 1
Ground truth answer: {}
Predicted answer: {}
Judegement:
```

Figure 19: Rule-based answer extraction strategy for OE questions.

**Prompt for Caption Generation**

```
Describe the fine-grained content of the image or figure, including
scenes, objects, relationships, and any text present.
```

Figure 20: Prompt template for caption generation.

| Physical Reasoning | Description |
|---|---|
| Physical Model Grounding Reasoning | This reasoning involves connecting the specific details of a problem description to fundamental physical concepts, laws, and idealized models. It's the process of identifying which area of physics is relevant and selecting the appropriate simplified representations that allow the problem to be analyzed using established physical principles and equations. Essentially, it translates a real-world or described scenario into a solvable physics framework. |
| Spatial Relation Reasoning | This focuses on understanding and manipulating the geometric and directional aspects of a physics problem. It involves visualizing the setup, determining the positions, orientations, distances, angles, and relative movements of objects. This often requires using coordinate systems, vectors (including resolving them into components), and geometric principles. |
| Multi-Formula Reasoning | This reasoning type is required when a problem cannot be solved using a single physics equation. It involves identifying multiple relevant formulas or principles and understanding how they interrelate. The process typically involves using the output of one formula as the input for another, or setting up and solving a system of simultaneous equations derived from different physical laws. |
| Implicit Condition Reasoning | This involves recognizing and utilizing information or constraints that are not explicitly stated in the problem text but are implied by the context, standard physics assumptions, or specific keywords. Examples include understanding that "starts from rest" means the initial velocity is zero, a "smooth" surface implies zero friction, a "light string" or "light pulley" means its mass is negligible, or that an object reaching its maximum height has a momentary vertical velocity of zero. |
| Numerical Reasoning | This reasoning refers to problems where solving requires the application of advanced mathematical methods beyond basic algebra and trigonometry. This includes techniques such as calculus, solving differential equations that model the system, vector calculus, Fourier analysis, linear algebra for complex systems, or other higher-level mathematical procedures necessary to manipulate the physical formulas and arrive at a solution. This applies when the mathematical technique itself is a core part of solving the physics, regardless of whether the final answer is purely numerical or symbolic. |
| Predictive Reasoning | This involves using established physical laws and the initial conditions of a system to forecast its future state or behavior. Based on the principles governing the situation, you calculate or deduce what will happen after a certain time or interaction. Examples include predicting the trajectory of a projectile, the final temperature of a mixture after thermal equilibrium is reached, or the velocity of objects after a collision. |

Table 6: Definitions of six physical reasoning categories in PHYX.

## Prompt for Reasoning Type Labeling(1)

You are an expert AI assistant specializing in analyzing physics problems.**

**Your Task:**
Your goal is to carefully read the provided physics problem and identify the **zero, one, or two MOST critical/dominant** reasoning types required to solve it, based on the definitions below. Your primary task is selection and prioritization. Assign a **maximum of two** labels per problem.

**Reasoning Type Definitions:**

1. **Physical Model Grounding Reasoning:**
   * Explanation: Connecting problem details to physical concepts, laws, and idealized models (e.g., point mass, frictionless surface, ideal gas). Translating the scenario into a physics framework.

2. **Spatial Relation Reasoning:**
   * Explanation: Understanding and manipulating geometric aspects (positions, angles, vectors, diagrams, coordinate systems).

3. **Multi-Formula Reasoning:**
   * Explanation: Requiring the combination or sequential use of multiple distinct physics formulas or principles to find the solution.

4. **Implicit Condition Reasoning:**
   * Explanation: Recognizing and using conditions not explicitly stated but implied by context or keywords (e.g., "starts from rest", "smooth surface", "maximum height").

5. **Numerical Reasoning:** (Revised Definition)
   * **Explanation:** Requiring advanced mathematical methods beyond basic algebra/trigonometry (e.g., calculus - integration/ differentiation, solving differential equations, Fourier analysis) as a core part of manipulating physical formulas. Do *not* select this for basic algebra or substitutions.

6. **Predictive Reasoning:**
   * Explanation: Using physical laws and initial conditions to forecast a future state or behavior (e.g., final velocity, trajectory, final temperature).

Figure 21: Prompt for reasoning type labeling (1).

## Prompt for Reasoning Type Labeling(2)

**Instructions:**

1. **Read and Analyze:** Carefully understand the problem and the likely steps/concepts needed for its solution.
2. **Identify Potential Types:** Determine which of the 6 reasoning types are involved in the solution process.
3. **Prioritize and Select:** From the potentially involved types, select **at most two** that are the **most critical, dominant, or uniquely challenging** aspects of solving this *particular* problem.
   * Think about what makes the problem non-trivial. Is it complex geometry? Combining multiple physics laws? Recognizing hidden conditions? Needing calculus?
   * If several types apply, choose the one or two that best represent the core difficulty or the essential nature of the solution process.
   * If only one type truly stands out as the most essential characteristic, list only that one.
   * If the problem is exceptionally simple or doesn't strongly fit any category as "most critical", output zero labels (`[]`).
4. **Output Format:** **CRITICAL:** Your entire response must consist *only* of a single Python-style list containing strings of the exact names for the selected zero, one, or two most critical reasoning types. Do **NOT** include any introductory text, explanations, labels, apologies, or any characters outside of the list itself.
   * **Correct Format Example (Two Types):** `["Reasoning Type A", "Reasoning Type B"]`
   * **Correct Format Example (One Type):** `["Reasoning Type C"]`
   * **Correct Format Example (Zero Types):** `[]`
   * **Incorrect Format Example:** `The most critical types are: ["Reasoning Type A", "Reasoning Type B"]`

**Example demonstrating the task (internal analysis, not part of the output):**

* **Problem:** "A 2 kg block, initially at rest on a frictionless horizontal surface, is pulled by a constant horizontal force of 10 N. What is its velocity after it has traveled 5 meters?"
* **(Internal Analysis:** Potential types involved: Grounding, Spatial (trivial here), Multi-Formula (F=ma then kinematics), Implicit ('at rest'), Predictive. Which are *most critical* (max 2)? Combining F=ma and kinematics (`Multi-Formula Reasoning`) is the core physics calculation. Recognizing 'at rest' (`Implicit Condition Reasoning`) is crucial for the setup. These seem most central.)*
* **Required Output for this Example:** `["Multi-Formula Reasoning", "Implicit Condition Reasoning"]`

**Now, analyze the following physics problem:**

--- Problem Start ---

Figure 22: Prompt for reasoning type labeling (2).

# H CASE STUDY

## List of Case Study Figures

| Domain | Correct | Visual Reasoning Error | Text Reasoning Error | Lack of Knowledge |
|---|---|---|---|---|
| Mechanics | 1, 2 | 3 | 4 | 5 |
| Electromagnetism | 6, 7 | 8 | 9 | 10 |
| Thermodynamics | 11, 12 | 13 | 14 | 15 |
| Wave/Acoustics | 16, 17 | 18 | 19 | 20 |
| Optics | 21, 22 | 23 | 24 | 25 |
| Modern Physics | 26, 27 | 28 | 29 | 30 |

Table 7: Table index of case study figures by domains with associated error categories.

# Mechanics ✅

**Description:**

A firefighter, a distance $d$ from a burning building, directs a stream of water from a fire hose at angle $\theta_i$ above the horizontal as shown in figure.

**Question:**

If the initial speed of the stream is $v_i$, at what height $h$ does the water strike the building?

**GPT-4o:**

To determine the height $h$ at which the water strikes the building, we will analyze the motion of the water as a projectile. The motion can be broken into horizontal and vertical components. Here's the step-by-step reasoning:

Step 1: Break the initial velocity into components

The initial velocity $v_i$ can be resolved into horizontal and vertical components: $$v_{ix} = v_i \cos\theta_i$$ $$v_{iy} = v_i \sin \theta_i$$

Step 2: Time of flight to reach the building

The horizontal distance $d$ is covered at a constant horizontal velocity $v_{ix}$. The time $t$ it takes for the water to reach the building is given by: $$d = v_{ix} t$$
Substitute $v_{ix} = v_i \cos \theta_i$: $$t = \frac{d}{v_i \cos \theta_i}$$

Step 3: Vertical displacement at time $t$

The vertical displacement $y$ of the water at time $t$ is determined by the kinematic equation:
$$y = v_{iy} t - \frac{1}{2} g t^2$$
Substitute $v_{iy} = v_i \sin \theta_i$ and $t = \frac{d}{v_i \cos \theta_i}$:
$$y = \left( v_i \sin \theta_i \right) \left( \frac{d}{v_i \cos \theta_i} \right) - \frac{1}{2} g \left( \frac{d}{v_i \cos \theta_i} \right)^2$$
 Simplify the terms:
$$y = \frac{d \tan \theta_i}{1} - \frac{g d^2}{2 v_i^2 \cos^2 \theta_i}$$

Step 4: Interpret the result

The height $h$ at which the water strikes the building is the vertical displacement $y$ when the water reaches the horizontal distance $d$. Thus:
$$h = d \tan \theta_i - \frac{g d^2}{2 v_i^2 \cos^2 \theta_i}$$

Final Answer:
The height $h$ at which the water strikes the building is:
$$h = d \tan \theta_i - \frac{g d^2}{2 v_i^2 \cos^2 \theta_i}$$

Where:
- $d$ is the horizontal distance to the building,
- $\theta_i$ is the angle of the water stream above the horizontal,
- $v_i$ is the initial speed of the water,
- $g$ is the acceleration due to gravity ($9.8 \, \text{m/s}^2$).

**Reasoning Type: Physical Model Grounding Reasoning , Spatial Relation Reasoning**

**Ground Truth: dtanθi−gd2/2vi2cos2θi**

Figure 1: A sample correct case of Mechanics.

# Mechanics 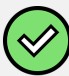

**Description:**

A landscape architect is designing an artificial waterfall in a city park. Water flows at a speed of 1.70 m/s as it exits the end of a horizontal channel located at the top of a vertical wall of height h=2.35 m. From there, the water falls freely into a pool below, as shown in the figure.

**Question:**

To sell her plan to the city council, the architect wants to build a model to standard scale, which is one-twelfth actual size. How fast should the water flow in the channel in the model?

**Options:**

A. 0.325m/s          B. 0.120m/s          C. 0.491m/s          D. 0.212m/s

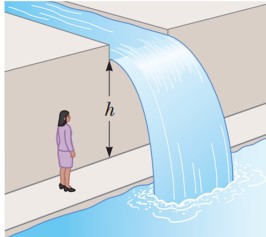

**GPT-4o:**

The speed of water in the model can be determined using the principle of similarity. The velocity in the model is scaled by the square root of the length scale factor. Since the model is one-twelfth the size of the actual waterfall, the length scale factor is $1/12$.

The velocity in the model is given by:
$$ v_{\text{model}} = v_{\text{actual}} \cdot \sqrt{\frac{\text{scale of model}}{\text{scale of actual}}} $$

Substituting the values:
$$ v_{\text{model}} = 1.70 \cdot \sqrt{\frac{1}{12}} = 1.70 \cdot \sqrt{0.0833} \approx 1.70 \cdot 0.2887 \approx 0.491 \, \text{m/s} $$

The correct answer is: C

**Reasoning Type: Physical Model Grounding Reasoning , Multi-Formula Reasoning**

**Ground Truth: C . 0.491m/s**

Figure 2: A sample correct case of Mechanics.

# Mechanics ⊗

**Description:**

We wrap a light, nonstretching cable around a solid cylinder, of mass 50 kg and diameter 0.120 m, that rotates in frictionless bearings about a stationary horizontal axis as shown in figure. We pull the free end of the cable with a constant 9.0 N force for a distance of 2.0 m; it turns the cylinder as it unwinds without slipping. The cylinder is initially at rest.

**Question:**

Find the final speed of the cable.

**GPT-4o:**

To solve this problem, we will use the principles of work and energy, as well as rotational dynamics. Here's the step-by-step reasoning:

**Step 1: Understand the system**
......

**Step 2: Work done by the force**
......

**Step 3: Rotational kinetic energy of the cylinder**
......

**Step 4: Relationship between linear and angular velocity**
......

**Step 5: Total kinetic energy**

The total kinetic energy of the system is the sum of the rotational kinetic energy of the cylinder and the translational kinetic energy of the cable:

$$K_{\text{total}} = K_{\text{rot}} + K_{\text{trans}},$$

where $K_{\text{trans}} = \frac{1}{2} m v^2$.
Substitute $K_{\text{rot}} = \frac{1}{2} I \omega^2$ and $\omega = \frac{v}{r}$:

$$K_{\text{rot}} = \frac{1}{2} I \left( \frac{v}{r} \right)^2 = \frac{1}{2} \left( 0.090 \right) \left( \frac{v}{0.060} \right)^2.$$

Simplify:

$$K_{\text{rot}} = \frac{1}{2} \cdot 0.090 \cdot \frac{v^2}{0.060^2} = \frac{0.090}{2 \cdot 0.060^2} v^2 = 12.5 v^2.$$

The translational kinetic energy is:

$$K_{\text{trans}} = \frac{1}{2} m v^2 = \frac{1}{2} (50) v^2 = 25 v^2.$$

Thus, the total kinetic energy is:

$$K_{\text{total}} = K_{\text{rot}} + K_{\text{trans}} = 12.5 v^2 + 25 v^2 = 37.5 v^2.$$

**Step 6: Work-energy principle**
......

**Final Answer:**
The final speed of the cable is approximately:

$$\boxed{0.69 \, \text{m/s}}.$$

**Reasoning Type: Multi-Formula Reasoning , Physical Model Grounding Reasoning**

**Error Type: Text Reasoning Errors**

**Ground Truth: 1.2 m/s**

Figure 3: A sample error case of Mechanics. Error category: Visual Reasoning Error

# Mechanics ⊗

### Description:

We want to slide a 12-kg crate up a 2.5-m-long ramp inclined at 30°. A worker, ignoring friction, calculates that he can do this by giving it an initial speed of 5.0 m/s at the bottom and letting it go. But friction is not negligible; the crate slides only 1.6 m up the ramp, stops, and slides back down.

### Question:

How fast is the crate moving when it reaches the bottom of the ramp?

### GPT-4o:

To solve this problem, we will use the principles of energy conservation, taking into account the work done by friction. Here's the step-by-step reasoning:

Step 1: Define the system and energy components
......

Step 2: Write the energy conservation equation
......

Step 3: Calculate the work done by friction**
The work done by friction is: $$ W_{\text{friction}} = f_k d $$
The frictional force $f_k$ is given by: $$ f_k = \mu_k N $$
Where:
- $\mu_k$: Coefficient of kinetic friction.
- $N = m g \cos \theta$: Normal force on the ramp.
The total distance traveled by the crate is:
$$ d = 1.6 \, \text{m (up)} + 1.6 \, \text{m (down)} = 3.2 \, \text{m}. $$
Thus: $$ W_{\text{friction}} = \mu_k m g \cos \theta \cdot d $$

Step 4: Solve for the final velocity
......

Step 5: Substitute known values
Given:
- $v_1 = 5.0 \, \text{m/s}$,
- $g = 9.8 \, \text{m/s}^2$,
- $\theta = 30^\circ$,
- $d = 3.2 \, \text{m}$,
- $\mu_k = \text{unknown (not provided)}$.
If $\mu_k$ is provided, substitute it into the equation to calculate $v_2$.

**Reasoning Type: Physical Model Grounding Reasoning , Multi-Formula Reasoning**

**Error Type: Text Reasoning Errors**

**Ground Truth: 2.5 m/s**

Figure 4: A sample error case of Mechanics. Error category: Text Reasoning Error

1674
1675
1676
1677
1678
1679
1680
1681
1682
1683
1684
1685
1686
1687
1688
1689
1690
1691
1692
1693
1694
1695
1696
1697
1698
1699
1700
1701
1702
1703
1704
1705
1706
1707
1708
1709
1710
1711
1712
1713
1714
1715
1716
1717
1718
1719
1720
1721
1722
1723
1724
1725
1726
1727

# Mechanics ✖

### Description:

A 500.0 g bird is flying horizontally at 2.25 m/s, not paying much attention, when it suddenly flies into a stationary vertical bar, hitting it 25.0 cm below the top as shown in figure. The bar is uniform, 0.750 m long, has a mass of 1.50 kg, and is hinged at its base.

### Question:

What is the angular velocity of the bar just as it reaches the ground?

### GPT-4o:

To solve this problem, we need to determine the angular velocity of the bar just as it reaches the ground. This involves analyzing the collision and the subsequent rotational motion of the bar. Here's the step-by-step solution:

Step 1: Conservation of Angular Momentum
The bird collides with the bar, imparting angular momentum to it. Since the hinge is frictionless, angular momentum about the hinge is conserved during the collision.
Angular Momentum Before Collision
The bird has linear momentum before the collision:
$$ p_{\text{bird}} = m_{\text{bird}} v_{\text{bird}} $$
...

The bird's angular momentum about the hinge is:
$$ L_{\text{bird}} = p_{\text{bird}} \cdot r $$
where $( r = 0.750 - 0.250 = 0.500 \, \text{m} )$ is the perpendicular distance from the hinge to the point of collision.
$$ L_{\text{bird}} = (1.125)(0.500) = 0.5625 \, \text{kg·m}^2/\text{s} $$
Angular Momentum After Collision
After the collision, the bar rotates about the hinge. Its angular momentum is:
$$ L_{\text{bar}} = I_{\text{bar}} \cdot \omega $$
where:
- $( I_{\text{bar}} )$ is the moment of inertia of the bar about the hinge,
- $( \omega )$ is the angular velocity of the bar.
...
Using conservation of angular momentum:
$$ L_{\text{bird}} = L_{\text{bar}} $$
$$ 0.5625 = (0.28125) \cdot \omega $$
Solve for $( \omega )$:
$$ \omega = \frac{0.5625}{0.28125} = 2.00 \, \text{rad/s} $$
Thus, the angular velocity of the bar immediately after the collision is:
$$ \omega = 2.00 \, \text{rad/s} $$

Step 2: Energy Conservation During Rotation
After the collision, the bar rotates about the hinge. As it falls, its potential energy is converted into rotational kinetic energy.
Initial Potential Energy
...
Solve for $( \omega )$:
$$ \omega^2 = \frac{2(5.5125)}{0.28125} = 39.2 $$
$$ \omega = \sqrt{39.2} = 6.26 \, \text{rad/s} $$

Final Answer:
The angular velocity of the bar just as it reaches the ground is:
$$ \boxed{6.26 \, \text{rad/s}} $$

### Reasoning Type: Multi-Formula Reasoning , Physical Model Grounding Reasoning

### Error Type: Lack of knowledge

### Ground Truth: 6.58 rad/s

Figure 5: A sample error case of Mechanics. Error category: Lack of Knowledge

# Electromagnetism ✅

**Description:**

In the slidewire, energy is dissipated in the circuit owing to its resistance. Let the resistance of the circuit (made up of the moving slidewire and the U-shaped conductor that connects the ends of the slidewire) at a given point in the slidewire's motion be R.

**Question:**

Find the rate at which energy is dissipated in the circuit.

**GPT-4o:**

To find the rate at which energy is dissipated in the circuit, we need to calculate the power dissipated due to the resistance $R$. The power dissipated in a resistor is given by:
$$P = \frac{\mathcal{E}^2}{R},$$
where $\mathcal{E}$ is the electromotive force (emf) induced in the circuit. Let's calculate $\mathcal{E}$ and then substitute it into the formula for power dissipation.

Step 1: Determine the induced emf ($\mathcal{E}$)
The emf induced in the circuit is due to the motion of the slidewire in the magnetic field $\mathbf{B}$. According to Faraday's law of electromagnetic induction, the emf is given by: $$\mathcal{E} = B L v,$$
where:
- $B$ is the magnitude of the uniform magnetic field (perpendicular to the plane of the circuit),
- $L$ is the length of the slidewire (the part of the circuit moving in the magnetic field),
- $v$ is the velocity of the slidewire (perpendicular to both $\mathbf{B}$ and $L$).

Step 2: Calculate the current in the circuit
The current $I$ in the circuit is related to the emf and the resistance $R$ by Ohm's law:
$$I = \frac{\mathcal{E}}{R}.$$
Substituting $\mathcal{E} = B L v$, we get: $$I = \frac{B L v}{R}.$$

Step 3: Calculate the power dissipated in the circuit
The power dissipated in the circuit due to the resistance $R$ is given by: $$P = I^2 R.$$
Substitute $I = \frac{B L v}{R}$ into this expression: $$P = \left( \frac{B L v}{R} \right)^2 R.$$
Simplify the expression: $$P = \frac{(B L v)^2}{R}.$$

Final Answer:
The rate at which energy is dissipated in the circuit is:
$$P = \frac{(B L v)^2}{R}.$$

This is the power dissipated in the circuit due to its resistance.

**Reasoning Type: Physical Model Grounding Reasoning, Multi-Formula Reasoning**

**Ground Truth:** $B^2 L^2 v^2/R$

Figure 6: A sample correct case of Electromagnetism.

# Electromagnetism ✓

### Description:

A sensitive electronic device with a resistance of R=175 $\Omega$ is to be connected to a source of emf (with negligible internal resistance) by a switch. The device is designed to operate with a current of 36 mA, but to avoid damage to the device, the current can rise to no more than 4.9mA in the first 58 $\mu$s after the switch is closed. An inductor is therefore connected in series with the device, as shown in the figure; the switch in question is S1.

### Question:

What is the R-L time constant $\tau$ ?

### Options:

A. 220$\mu$H        B. 390$\mu$H        C. 750$\mu$H        D. 280$\mu$H

Closing switch $S_1$ connects the *R-L* combination in series with a source of emf $\mathcal{E}$.

Closing switch $S_2$ while opening switch $S_1$ disconnnects the combination from the source.

### GPT-4o:

The $R\text{-}L$ time constant $\tau$ is given by:
$$\tau = \frac{L}{R}.$$

The current in an $R\text{-}L$ circuit grows according to:
$$i(t) = I_{\text{max}} \left( 1 - e^{-t/\tau} \right),$$
where $I_{\text{max}} = 36\ \mathrm{mA}$, $i(t) = 4.9\ \mathrm{mA}$, and $t = 58\ \mu\mathrm{s}$.

Rearranging for $\tau$:
$$e^{-t/\tau} = 1 - \frac{i(t)}{I_{\text{max}}}.$$

Substitute the values:
$$e^{-58 \times 10^{-6}/\tau} = 1 - \frac{4.9 \times 10^{-3}}{36 \times 10^{-3}} = 1 - 0.1361 = 0.8639.$$

Take the natural logarithm:
$$-\frac{58 \times 10^{-6}}{\tau} = \ln(0.8639).$$
$$\tau = -\frac{58 \times 10^{-6}}{\ln(0.8639)}.$$
$$\ln(0.8639) \approx -0.146.$$
$$\tau = \frac{58 \times 10^{-6}}{0.146} \approx 397 \times 10^{-6}\ \mathrm{s}.$$

Now, calculate $L$:
$$L = \tau R = (397 \times 10^{-6})(175) \approx 69.5 \times 10^{-3}\ \mathrm{H} = 390\ \mu\mathrm{H}.$$

Correct option: B

### Reasoning Type: Multi-Formula Reasoning , Numerical Reasoning

### Ground Truth: B . 390μH

Figure 7: A sample correct case of Electromagnetism.

# Electromagnetism ⊗

**Description:**

In the circuit, the switch S has been open for a long time and is suddenly closed. Neither the battery nor the inductors have any appreciable resistance.

**Question:**

What does the voltmeter read 0.115 ms after S is closed?

**GPT-4o:**

To solve this problem, we need to analyze the circuit and determine the voltage across the voltmeter $V$ 0.115 ms after the switch $S$ is closed. Here's the step-by-step reasoning:

---

### Step 1: Understand the circuit
- The circuit consists of a 20.0 V battery, resistors, and inductors.
- The switch $S$ has been open for a long time, so initially, no current flows through the circuit.
- When $S$ is closed, current starts flowing, and the inductors oppose the change in current due to their inductance.

---

Step 2: Inductors' behavior
...

Step 3: Equivalent inductance of the parallel inductors
...

Step 4: Time constant of the RL circuit
...

Step 5: Current in the circuit as a function of time
...

Step 6: Voltage across the parallel inductors
...
Now calculate $V$:
$$V = L_{\text{eq}} \frac{dI}{dt}$$
Substitute $L_{\text{eq}} = 4.87 \, \text{mH} = 4.87 \times 10^{-3} \, \text{H}$ and $\frac{dI}{dt} = 699 \, \text{A/s}$:
$$V = 4.87 \times 10^{-3} \cdot 699 \approx 3.41 \, \text{V}$$

---

Final Answer:
The voltmeter reads approximately **3.41 V** 0.115 ms after the switch is closed.

**Reasoning Type: Physical Model Grounding Reasoning , Numerical Reasoning**

**Error Type: Visual Reasoning Error**

**Ground Truth: 9.0 V**

Figure 8: A sample error case of Electromagnetism. Error category: Visual Reasoning Error

# Electromagnetism ⊗

**Description:**

Figure shows an electric dipole in a uniform electric field of magnitude 5.0 * 10^5 N/C that is directed parallel to the plane of the figure. The charges are pm 1.6 * 10^-19 C; both lie in the plane and are separated by 0.125 * 10^-9m.

**Question:**

Find the magnitude of the torque.

**GPT-4o:**

To calculate the torque ($\tau$) acting on the electric dipole in the uniform electric field, we use the formula:
$$\tau = pE \sin \theta$$
where:
- $p$ is the dipole moment,
- $E$ is the magnitude of the electric field,
- $\theta$ is the angle between the dipole moment ($\vec{p}$) and the electric field ($\vec{E}$).

Step 1: Calculate the dipole moment ($p$)
The dipole moment is given by:
$$p = qd$$
where:
- $q = 1.6 \times 10^{-19}\ \text{C}$ is the charge,
- $d = 0.125\ \text{m}$ is the separation between the charges.

Substitute the values:
$$p = (1.6 \times 10^{-19})(0.125)$$ $$p = 2.0 \times 10^{-20}\ \text{C·m}$$

Step 2: Use the torque formula
...

Final Answer:
The magnitude of the torque is:
$$\tau \approx 5.7 \times 10^{-15}\ \text{N·m}$$

**Reasoning Type:** Physical Model Grounding Reasoning , Spatial Relation Reasoning

**Error Type:** Text Reasoning Errors

**Ground Truth:** 5.7 * 10^{-24} \, \text{N} \cdot \text{m}

Figure 9: A sample error case of Electromagnetism. Error category: Text Reasoning Error

# Electromagnetism ⊗

**Description:**

Point charges q_1 = +12 nC and q_2 = -12 nC are 0.100 m apart. (Such pairs of point charges with equal magnitude and opposite sign are called electric dipoles.

**Question:**

Compute the total field at point c.

**GPT-4o:**

To compute the total electric field at point $c$ due to the charges $q_1$ and $q_2$, we will calculate the contributions of the electric fields from each charge and then combine them vectorially. Here's the step-by-step reasoning:

---

Step 1: Understand the setup
- $q_1 = +12 \, \text{nC}$ (positive charge).
- $q_2 = -12 \, \text{nC}$ (negative charge).
- Distance between $q_1$ and $q_2$: $0.100 \, \text{m}$.
- Point $c$ is equidistant from both charges, at a distance of $r = 0.130 \, \text{m}$ from each charge.
- The electric field at $c$ due to each charge is directed along the line joining the charge and point $c$.

---

Step 2: Electric field due to a point charge
The electric field due to a point charge is given by:
$$E = \frac{k |q|}{r^2},$$
where:
- $k = 8.99 \times 10^9 \, \text{N·m}^2/\text{C}^2$ (Coulomb's constant),
- $q$ is the magnitude of the charge,
- $r$ is the distance from the charge to the point where the field is being calculated.

---

Step 3: Calculate the magnitude of the electric field at $c$ due to $q_1$ and $q_2$**
...

Final Answer:
The total electric field at point $c$ is:
$$\boxed{11.8 \, \text{N/C} \, \text{(upward along the vertical axis)}}.$$

**Reasoning Type: Physical Model Grounding Reasoning, Spatial Relation Reasoning**

**Error Type: Lack of Knowledge**

**Ground Truth: 5.7 * 10^{-24} N/C**

Figure 10: A sample error case of Electromagnetism. Error category: Lack of Knowledge

# Thermodynamics ✅

**Description:**

A partition divides a thermally insulated box into two compartments, each of volume V(figure). Initially, one compartment contains n moles of an ideal gas at temperature T, and the other compartment is evacuated. We break the partition and the gas expands, filling both compartments.

**Question:**

What is the entropy change in this free-expansion process?

**GPT-4o:**

To calculate the root-mean-square (rms) speed $v_{\mathrm{rms}}$ of the molecules in the gas, we use the formula: $$ v_{\mathrm{rms}} = \sqrt{\frac{1}{N} \sum_{i=1}^N v_i^2} $$
 where:
- $N$ is the number of molecules,
- $v_i$ is the speed of the $i$-th molecule, and
- $v_i^2 = v_{ix}^2 + v_{iy}^2$, where $v_{ix}$ and $v_{iy}$ are the $x$- and $y$-components of the velocity of the $i$-th molecule.

Step 1: Extract the velocity components for each molecule
From the diagram, the velocity components for each molecule are:
1. Molecule 1: $\vec{v}_1 = 10\hat{i} - 10\hat{j}$   $v_{1x} = 10, \, v_{1y} = -10$
2. Molecule 2: $\vec{v}_2 = 2\hat{i} + 15\hat{j}$   $v_{2x} = 2, \, v_{2y} = 15$
......
6. Molecule 6: $\vec{v}_6 = 0\hat{i} - 14\hat{j}$   $v_{6x} = 0, \, v_{6y} = -14$

Step 2: Calculate the speed $v_i$ for each molecule
The speed of each molecule is given by: $$ v_i = \sqrt{v_{ix}^2 + v_{iy}^2} $$
1. Molecule 1:  $$ v_1 = \sqrt{10^2 + (-10)^2} = \sqrt{100 + 100} = \sqrt{200} = 10\sqrt{2} $$
2. Molecule 2:  $$ v_2 = \sqrt{2^2 + 15^2} = \sqrt{4 + 225} = \sqrt{229} $$
......
6. Molecule 6:  $$ v_6 = \sqrt{0^2 + (-14)^2} = \sqrt{0 + 196} = 14 $$

Step 3: Calculate $v_i^2$ for each molecule
We now square the speeds:
1. $v_1^2 = (10\sqrt{2})^2 = 200$
2. $v_2^2 = (\sqrt{229})^2 = 229$
......
6. $v_6^2 = 14^2 = 196$

Step 4: Calculate the mean of $v_i^2$
The mean of $v_i^2$ is:
$$ \text{Mean} = \frac{1}{N} \sum_{i=1}^N v_i^2 = \frac{1}{6} (200 + 229 + 100 + 104 + 61 + 196) $$
First, calculate the sum: $$ 200 + 229 + 100 + 104 + 61 + 196 = 890 $$
Now divide by $N = 6$: $$ \text{Mean} = \frac{890}{6} \approx 148.33 $$

Step 5: Calculate $v_{\mathrm{rms}}$
Finally, take the square root of the mean:
$$ v_{\mathrm{rms}} = \sqrt{148.33} \approx 12.18 $$

Final Answer:
The rms speed is approximately: $$ v_{\mathrm{rms}} \approx 12.2 \, \text{units} $$

**Reasoning Type: Physical Model Grounding Reasoning , Numerical Reasoning**

**Ground Truth: 12.2m/s**

Figure 11: A sample correct case of Thermodynamics.

2052
2053
2054
2055
2056
2057
2058
2059
2060
2061
2062
2063
2064
2065
2066
2067
2068
2069
2070
2071
2072
2073
2074
2075
2076
2077
2078
2079
2080
2081
2082
2083
2084
2085
2086
2087
2088
2089
2090
2091
2092
2093
2094
2095
2096
2097
2098
2099
2100
2101
2102
2103
2104
2105

# Thermodynamics 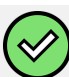

**Description:**

Two piston/cylinder arrangements, A and B, have their gas chambers connected by a pipe, as shown in the figure. The cross-sectional areas of the pistons are $A_A$=75 cm$^2$ and $A_B$=25 cm$^2$, with the piston mass in arrangement A being $m_A$=25 kg. Assume an outside pressure of 100kPa and standard gravitational acceleration.

**Question:**

Find the mass $m_B$ such that none of the pistons rests on the bottom.

**Options:**

A. 490kg          B. 8.33kg          C. 154kg          D. 10.2kg

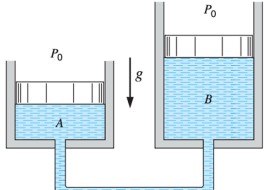

**GPT-4o:**

To solve this, we use the principle of pressure equilibrium. The pressure in both chambers must be equal since they are connected by a pipe.

Step 1: Pressure in chamber A
The pressure in chamber A is given by:
$$ P_A = P_0 + \frac{m_A g}{A_A} $$
Substitute the values:
$$ P_A = 100,000 + \frac{25 \cdot 9.81}{0.0075} = 100,000 + 32,700 = 132,700 \, \text{Pa} $$

Step 2: Pressure in chamber B
The pressure in chamber B is given by:
$$ P_B = P_0 + \frac{m_B g}{A_B} $$
Since $P_A = P_B$, we equate the two pressures:
$$ 132,700 = 100,000 + \frac{m_B \cdot 9.81}{0.0025} $$

Step 3: Solve for $m_B$
Rearrange to isolate $m_B$:
$$ \frac{m_B \cdot 9.81}{0.0025} = 32,700 $$
$$ m_B = \frac{32,700 \cdot 0.0025}{9.81} = 8.33 \, \text{kg} $$

Final Answer: B

**Reasoning Type: Physical Model Grounding Reasoning , Multi-Formula Reasoning**

**Ground Truth: B . 8.33kg**

Figure 12: A sample correct case of Thermodynamics.

# Thermodynamics ⊗

### Description:

The graph in figure shows a PV-diagram of the air in a human lung when a person is inhaling and then exhaling a deep breath. Such graphs, obtained in clinical practice, are normally somewhat curved, but we have modeled one as a set of straight lines of the same general shape. (Important: The pressure shown is the gauge pressure, not the absolute pressure.) The process illustrated here is somewhat different from those we have been studying, because the pressure change is due to changes in the amount of gas in the lung, not to temperature changes.

### Question:

If the temperature of the air in the lung remains a reasonable 20°C, what is the maximum number of moles in this person's lung during a breath?

### GPT-4o:

To determine the maximum number of moles of air in the lung during a breath, we can use the ideal gas law: $$pV = nRT$$

Where:
- $p$ is the pressure (in pascals, Pa),
- $V$ is the volume (in cubic meters, m³),
- $n$ is the number of moles,
- $R$ is the ideal gas constant ($R = 8.314 \, \text{J/mol·K}$),
- $T$ is the temperature (in kelvins, K).

Step 1: Identify the maximum pressure and volume
From the graph:
- The maximum **gauge pressure** is $p_{\text{gauge}} = 12.0 \, \text{mmHg}$.
- The maximum volume is $V = 1.5 \, \text{L}$.

Step 2: Convert pressure and volume to SI units
......

Step 3: Convert temperature to kelvins
......

Step 4: Solve for the number of moles
......

Final Answer:
The maximum number of moles of air in the lung during a breath is approximately:
$$\boxed{0.0634 \, \text{mol}}$$

### Reasoning Type: Physical Model Grounding Reasoning , Multi-Formula Reasoning

### Error Type: Visual Reasoning Errors

### Ground Truth: 0.059mol

Figure 13: A sample error case of Thermodynamics. Error category: Visual Reasoning Error

# Thermodynamics ⊗

### Description:

A large tank of water has a hose connected to it. The tank is sealed at the top and has compressed air between the water surface and the top. When the water height h has the value 3.50 m, the absolute pressure p of the compressed air is 4.20 × 10^5 Pa. Assume that the air above the water expands at constant temperature, and take the atmospheric pressure to be 1.00 × 10^5 Pa.

### Question:

As water flows out of the tank, h decreases. At what value of h does the flow stop?

**4.00 m**

**1.00 m**

### GPT-4o:

To determine the value of $( h )$ at which the flow stops, we need to analyze the system step by step. The flow stops when the pressure at the outlet of the tank equals the atmospheric pressure. Here's the reasoning:

Step 1: Understand the system
......

Step 2: Pressure at the outlet
......

Step 3: Relating the air pressure $( p )$ to the water height $( h )$
......

Step 4: Solve for $( h )$ when the flow stops
Substitute $( p = \frac{(4.20 \times 10^5) (0.50)}{4.00 - h} )$ into the condition $( p + \rho g h = p_{\text{atm}} )$:
$$\frac{(4.20 \times 10^5) (0.50)}{4.00 - h} + (1000)(9.8) h = 1.00 \times 10^5.$$

Simplify:
$$\frac{2.10 \times 10^5}{4.00 - h} + 9800 h = 1.00 \times 10^5.$$

Rearrange:
$$\frac{2.10 \times 10^5}{4.00 - h} = 1.00 \times 10^5 - 9800 h.$$

......

Step 5: Solve the quadratic equation
......

Final Answer:
The flow stops when $( h = 1.54 , \text{m} )$.

### Reasoning Type: Multi-Formula Reasoning , Physical Model Grounding Reasoning

### Error Type: Text Reasoning Errors

### Ground Truth: 1.74m

Figure 14: A sample error case of Thermodynamics. Error category: Text Reasoning Error

# Thermodynamics 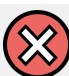

### Description:

A long rod, insulated to prevent heat loss along its sides, is in perfect thermal contact with boiling water (at atmospheric pressure) at one end and with an ice–water mixture at the other. The rod consists of a 1.00 m section of copper (one end in boiling water) joined end to end to a length $L_2$ of steel (one end in the ice–water mixture). Both sections of the rod have cross-sectional areas of 4.0 cm^2. The temperature of the copper–steel junction is $65.0^\circ\text{C}$ after a steady state has been set up.

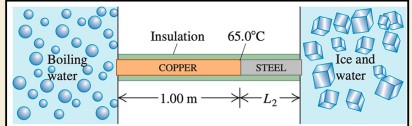

### Question:

What is the length $L_2$ of the steel section?

### GPT-4o:

To solve for the length $L_2$ of the steel section, we use the concept of steady-state heat conduction. In steady-state conditions, the heat flow rate through the copper section is equal to the heat flow rate through the steel section. This is because no heat is lost along the sides of the rod, and the system is in thermal equilibrium.

...

Step 1: Write the heat flow rate for copper and steel
For the copper section:
$$\frac{Q}{t} = \frac{k_{\text{Cu}} A (T_{\text{boiling}} - T_{\text{junction}})}{L_{\text{Cu}}},$$
...

Step 2: Set the heat flow rates equal
...

Step 4: Substitute known values
- $k_{\text{Cu}} = 390\,\text{W/m·K}$,
- $k_{\text{steel}} = 46\,\text{W/m·K}$,
- $T_{\text{boiling}} = 100.0^\circ\text{C}$,
- $T_{\text{junction}} = 65.0^\circ\text{C}$,
- $T_{\text{ice}} = 0.0^\circ\text{C}$,
- $L_{\text{Cu}} = 1.00\,\text{m}$.

Substitute these values into the equation:
...

Step 5: Perform the calculations
...

Final Answer:
The length of the steel section is:
$$\boxed{0.219\,\text{m}}.$$

**Reasoning Type: Physical Model Grounding Reasoning , Multi-Formula Reasoning**

**Error Type: Lack of Knowledge**

**Ground Truth: 5.7 * 10^{-24} N/C**

Figure 15: A sample error case of Thermodynamics. Error category: Lack of Knowledge

# Wave/Acoustics

**Description:**

A directional loudspeaker directs a sound wave of wavelength λ at a wall (figure)

**Question:**

At what distances from the wall could you stand and hear no sound at all?

**GPT-4o:**

To determine the distances from the wall where no sound is heard, we need to analyze the interference of the sound waves. Here's the step-by-step reasoning:

Step 1: Understand the setup
- A speaker emits sound waves toward a wall.
- The sound waves reflect off the wall and interfere with the incoming waves.
- At certain points, destructive interference occurs, resulting in no sound being heard.

Step 2: Condition for destructive interference
Destructive interference occurs when the path difference between the incident wave and the reflected wave is an odd multiple of half the wavelength ($\lambda/2$): $$\text{Path difference} = (2m + 1) \frac{\lambda}{2}, \quad m = 0, 1, 2, \dots$$
Here, $m$ is an integer representing the order of the interference.

Step 3: Path difference in terms of distance from the wall
- Let the distance from the wall be $d$.
- The sound wave travels a distance $d$ to the wall and another distance $d$ back after reflection.
- The total path difference is $2d$.
For destructive interference: $$2d = (2m + 1) \frac{\lambda}{2}$$

Step 4: Solve for $d$
Rearranging the equation for $d$: $$d = \frac{(2m + 1) \lambda}{4}, \quad m = 0, 1, 2, \dots$$
This gives the distances from the wall where destructive interference occurs.

Step 5: List the distances
For different values of $m$:
- $m = 0$: $d = \frac{\lambda}{4}$
- $m = 1$: $d = \frac{3\lambda}{4}$
- $m = 2$: $d = \frac{5\lambda}{4}$
- $m = 3$: $d = \frac{7\lambda}{4}$
- And so on...

Final Answer:
The distances from the wall where no sound is heard are:
$$d = \frac{\lambda}{4}, \frac{3\lambda}{4}, \frac{5\lambda}{4}, \frac{7\lambda}{4}, \dots$$

**Reasoning Type: Physical Model Grounding Reasoning , Spatial Relation Reasoning**

**Ground Truth: λ/4, 3λ/4, 5λ/4**

Figure 16: A sample correct case of Wave/Acoustics.

# Wave/Acoustics

**Description:**

Small speakers A and B are driven in phase at 725Hz by the same audio oscillator. Both speakers start out 4.50m from the listener, but speaker A is slowly moved away.

**Question:**

At what distance d will the sound from the speakers first produce destructive interference at the listener's location?

**Options:**

A. 0.237m                B. 0.336m                C. 0.569m                D. 0.414m

**GPT-4o:**

To determine the distance $d$ at which destructive interference first occurs, we use the condition for destructive interference:
$$\Delta L = \frac{\lambda}{2},$$
where $\Delta L$ is the path difference between the two speakers, and $\lambda$ is the wavelength of the sound.

Step 1: Calculate the wavelength $\lambda$
The speed of sound in air is approximately $v = 343 \, \text{m/s}$, and the frequency of the sound is $f = 725 \, \text{Hz}$. The wavelength is given by:
$$\lambda = \frac{v}{f} = \frac{343}{725} \approx 0.473 \, \text{m}.$$

Step 2: Path difference for destructive interference
For the first instance of destructive interference, the path difference $\Delta L$ is:
$$\Delta L = \frac{\lambda}{2} = \frac{0.473}{2} \approx 0.237 \, \text{m}.$$

Step 3: Relating $\Delta L$ to $d$
The path difference $\Delta L$ is caused by moving speaker $A$ a distance $d$ farther from the listener. Thus: $$\Delta L = d.$$
So: $$d = 0.237 \, \text{m}.$$

Final Answer: **A**

**Reasoning Type: Physical Model Grounding Reasoning , Spatial Relation Reasoning**

**Ground Truth: A . 0.237m**

Figure 17: A sample correct case of Wave/Acoustics.
Back to List of Figures | Back to Table Index

# Wave/Acoustics

### Description:

Very finely ground particles of cork are sprinkled in the bottom of the tube before the piston is inserted. As the vibrating piston is slowly moved forward, there are a few positions that cause the cork particles to collect in small, regularly spaced piles along the bottom. Figure shows an experiment in which the tube is filled with pure oxygen and the piston is driven at 400 Hz.

Piston · Glass tube · Piles of cork particles · 400 Hz · 123 cm

### Question:

What is the speed of sound in oxygen?

### GPT-4o:

To determine the speed of sound in oxygen using the given setup, we will analyze the standing wave pattern formed in the Kundt's tube. Here's the step-by-step reasoning:

Step 1: Understand the standing wave pattern
......

Step 2: Relationship between the wavelength and the distance between nodes
......

Step 3: Determine the wavelength
- From the diagram, the total length of the tube is $123 \, \text{cm} = 1.23 \, \text{m}$.
- The cork particles form regularly spaced piles, indicating the positions of the nodes.
- Count the number of node-to-node segments in the tube. From the diagram, there are 5 segments.
- The total length of the tube corresponds to 5 half-wavelengths:
$$5 \cdot \frac{\lambda}{2} = 1.23 \, \text{m}$$
- Solve for $\lambda$ (the wavelength):
$$\lambda = \frac{2 \cdot 1.23}{5} = 0.492 \, \text{m}$$
Step 4: Use the wave equation to find the speed of sound
......

Final Answer:
The speed of sound in oxygen is approximately:
$$\boxed{196.8 \, \text{m/s}}$$

**Reasoning Type: Physical Model Grounding Reasoning , Multi-Formula Reasoning**

**Error Type: Visual Reasoning Errors**

**Ground Truth: 1.2 m/s**

Figure 18: A sample error case of Wave/Acoustics. Error category: Visual Reasoning Error

2430
2431
2432
2433
2434
2435
2436
2437
2438
2439
2440
2441
2442
2443
2444
2445
2446
2447
2448
2449
2450
2451
2452
2453
2454
2455
2456
2457
2458
2459
2460
2461
2462
2463
2464
2465
2466
2467
2468
2469
2470
2471
2472
2473
2474
2475
2476
2477
2478
2479
2480
2481
2482
2483

# Wave/Acoustics 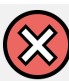

**Description:**
Your team is designing a landing aid for pilots in rain or fog by placing two radio transmitters 50 m apart on either side of a runway. These transmitters emit the same frequency but are out of phase, creating a nodal line down the runway center. The pilot hears nothing when aligned, but hears a beep if off-center. For accurate guidance, the first intensity maxima should appear 60 m from the nodal line at a distance of 3.0 km.

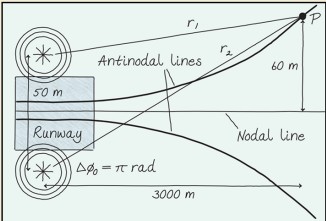

**Question:**
What frequency should you specify for the transmitters?

**GPT-4o:**

To determine the frequency of the transmitters, we need to analyze the interference pattern created by the two sources. The goal is to ensure that the first antinodal line (intensity maxima) is $60 \, \mathrm{m}$ away from the nodal line at a distance of $3.0 \, \mathrm{km}$ from the transmitters.

Step 1: Geometry of the problem
The two transmitters are separated by a distance $d = 50 \, \mathrm{m}$. The nodal line is the line of destructive interference, where the path difference between the two sources is $\Delta r = (m + \frac{1}{2}) \lambda$, with $m = 0, 1, 2, \dots$. The first antinodal line corresponds to constructive interference, where the path difference is $\Delta r = m \lambda$, with $m = 1$.

At a distance $L = 3000 \, \mathrm{m}$ from the transmitters, the first antinodal line is $y = 60 \, \mathrm{m}$ away from the nodal line. This forms a right triangle with the base $L$ and height $y$.

Step 2: Path difference for the first antinodal line
...

Step 3: Wavelength of the radio waves
For the first antinodal line, the path difference $\Delta r$ corresponds to one wavelength:
$$\Delta r = \lambda.$$
Thus: $$\lambda = 1.0 \, \mathrm{m}.$$

Step 4: Frequency of the radio waves
The frequency $f$ of the radio waves is related to the wavelength $\lambda$ and the speed of light $c$ by: $$f = \frac{c}{\lambda}.$$
Using $c = 3.0 \times 10^8 \, \mathrm{m/s}$ and $\lambda = 1.0 \, \mathrm{m}$:
$$f = \frac{3.0 \times 10^8}{1.0} = 3.0 \times 10^8 \, \mathrm{Hz}.$$

Final Answer:
The frequency of the transmitters should be: $$\boxed{300 \, \mathrm{MHz}}$$

**Reasoning Type: Spatial Relation Reasoning , Multi-Formula Reasoning**

**Error Type: Text Reasoning Errors**

**Ground Truth: 150 MHz**

Figure 19: A sample error case of Wave/Acoustics. Error category: Text Reasoning Error

# Wave/Acoustics ❌

**Description:**

Two loudspeakers are placed 3.00 m apart, as shown in figure. They emit 474 Hz sounds, in phase. A microphone is placed 3.20 m distant from a point midway between the two speakers, where an intensity maximum is recorded.

**Question:**

How far must the microphone be moved to the right to find the first intensity minimum?

**GPT-4o:**

To solve this problem, we need to determine the distance the microphone must be moved to the right to encounter the first intensity minimum. This involves understanding the conditions for destructive interference of sound waves.

Step 1: Conditions for Destructive Interference
......

Step 2: Wavelength of the Sound
......

Step 3: Geometry of the Problem
......

Step 4: Step 4: Approximation for Small $\( x \)$
For small $\( x \)$, we can use a first-order approximation for the square root:
$\[\sqrt{a^2 + b^2} \approx a + \frac{b^2}{2a}, \quad \text{if } b \ll a.\]$
Here, $\( a = 3.20 \, \mathrm{m} \)$ and $\( b = 1.50 \pm x \)$. Applying this approximation:$\[d\_2 \approx 3.20 + \frac{(1.50 + x)^2}{2(3.20)},\]$
......

Step 5: Solve for $\( x \)$
......

Final Answer:
The speed of the waves in the coffee is approximately:
$\[\boxed{0.08 \, \mathrm{m/s}}\]$

**Reasoning Type:** Physical Model Grounding Reasoning , Spatial Relation Reasoning

**Error Type:** Lack of Knowledge

**Ground Truth:** 0.429m

Figure 20: A sample error case of Wave/Acoustics. Error category: Lack of Knowledge

# Optics

**Description:**

A ray of light travels from air into another medium, making an angle of θ1=45.0° with the normal as in figure.

**Question:**

Find the angle of refraction θ2, if the second medium is water.

**GPT-4o:**

To solve this problem, we will use **Snell's Law**, which relates the angles of incidence and refraction to the refractive indices of the two media: $$ n_1 \sin \theta_1 = n_2 \sin \theta_2 $$
Where:
- $ n_1 $ is the refractive index of the first medium (air),
- $ n_2 $ is the refractive index of the second medium (water),
- $ \theta_1 $ is the angle of incidence,
- $ \theta_2 $ is the angle of refraction.

Step 1: Assign known values
- The refractive index of air is approximately $ n_1 = 1.00 $,
- The refractive index of water is $ n_2 = 1.33 $,
- The angle of incidence is $ \theta_1 = 45.0^\circ $.

Step 2: Apply Snell's Law
Substitute the known values into Snell's Law: $$ 1.00 \cdot \sin(45.0^\circ) = 1.33 \cdot \sin(\theta_2) $$

Step 3: Simplify the equation
The sine of $ 45.0^\circ $ is: $$ \sin(45.0^\circ) = \frac{\sqrt{2}}{2} \approx 0.707 $$
Substitute this value into the equation: $$ 1.00 \cdot 0.707 = 1.33 \cdot \sin(\theta_2) $$ $$ 0.707 = 1.33 \cdot \sin(\theta_2) $$

Step 4: Solve for $ \sin(\theta_2) $
Divide both sides of the equation by $ 1.33 $: $$ \sin(\theta_2) = \frac{0.707}{1.33} $$ $$ \sin(\theta_2) \approx 0.531 $$

Step 5: Find $ \theta_2 $
Take the inverse sine ($ \arcsin $) of $ 0.531 $ to find $ \theta_2 $: $$ \theta_2 = \arcsin(0.531) $$
Using a calculator: $$ \theta_2 \approx 32.0^\circ $$

Final Answer:
The angle of refraction is: $$ \boxed{\theta_2 \approx 32.0^\circ} $$

**Reasoning Type: Physical Model Grounding Reasoning , Spatial Relation Reasoning**

**Ground Truth: 32.0°**

Figure 21: A sample correct case of Optics.

# Optics

✓

**Description:**

Assume a transparent rod with diameter d=2.00 μm and an index of refraction of 1.36.

**Question:**

Determine the maximum angle θ at which light rays can be incident on the end of the rod (as shown in the figure) such that they undergo total internal reflection along the walls of the rod.

**Options:**

A. 72.2°          B. 65.4°          C. 67.2°          D. 60.0°

**GPT-4o:**

To determine the maximum angle $\theta$, we use the concept of total internal reflection. The critical angle $\theta_c$ is given by: $$\sin \theta_c = \frac{n_2}{n_1}$$ where $n_1 = 1.36$ (index of refraction of the rod) and $n_2 = 1.00$ (index of refraction of air).

Solving for $\theta_c$:
$$\sin \theta_c = \frac{1.00}{1.36} \implies \theta_c = \arcsin\left(\frac{1.00}{1.36}\right) \approx 47.8^\circ$$

The maximum angle $\theta$ for total internal reflection is related to the critical angle by:
$$\theta = 90^\circ - \theta_c$$

Substituting $\theta_c = 47.8^\circ$:
$$\theta = 90^\circ - 47.8^\circ = 42.2^\circ$$

However, this is the angle relative to the normal. For the angle relative to the axis of the rod, we use:
$$\theta = 90^\circ - 42.2^\circ = 67.2^\circ$$

Thus, the correct answer is: C

**Reasoning Type: Physical Model Grounding Reasoning , Spatial Relation Reasoning**

**Ground Truth: C . 67.2°**

Figure 22: A sample correct case of Optics.

# Optics

**Description:**

When the light ray illustrated in figure passes through the glass block of index of refraction n = 1.50, it is shifted laterally by the distance d."

**Question:**

Find the time interval required for the light to pass through the glass block.

**GPT-4o:**

To solve this problem, we will use Snell's Law and the geometry of the prism. The steps are as follows:

Step 1: Understand the geometry of the prism and the light path
......

Step 2: Relate the angles inside the prism
- Inside the prism, the beam travels at an angle of $45^\circ$ with respect to the normal to the hypotenuse. This is because the prism is isosceles, and the beam enters perpendicular to one leg.
- At the hypotenuse, the beam refracts and exits the prism. Let the angle of refraction (outside the prism) be $r$, measured with respect to the normal to the hypotenuse.

From the geometry of the problem:
$$\theta = 45^\circ - r$$
Substitute $\theta = 15.0^\circ$:
$$15.0^\circ = 45^\circ - r$$
$$r = 30.0^\circ$$

Step 3: Apply Snell's Law at the hypotenuse
......

Step 4: Solve for $n$
......

Final Answer:
The index of refraction of the prism is:
$$\boxed{1.414}$$

**Reasoning Type: Physical Model Grounding Reasoning , Spatial Relation Reasoning**

**Error Type:** Visual Reasoning Errors

**Ground Truth: 1.2m/s**

Figure 23: A sample error case of Optics. Error category: Visual Reasoning Error

# Optics

❌

**Description:**

When the light ray illustrated in figure passes through the glass block of index of refraction n = 1.50, it is shifted laterally by the distance d.

**Question:**

Find the time interval required for the light to pass through the glass block.

30.0°

2.00 cm

d

**GPT-4o:**

To find the time interval required for the light to pass through the glass block, we need to calculate the time it takes for the light to travel the distance it covers inside the block. Here's the step-by-step reasoning:

Step 1: Understand the problem
......

Step 2: Speed of light in the glass
......

Step 3: Distance traveled by the light in the glass
The light ray travels diagonally through the glass block. The thickness of the block is \( 2.00 \, \text{cm} = 0.0200 \, \text{m} \), and the light travels at an angle of \( 30^\circ \) with respect to the normal.
The actual distance \( L \) traveled by the light inside the block is the hypotenuse of a right triangle, where the thickness of the block is the adjacent side. Using trigonometry:
\[L = \frac{\text{thickness}}{\cos(30^\circ)}.\]
Substitute the values:
\[L = \frac{0.0200}{\cos(30^\circ)}.\]
...

Step 4: Time taken to travel through the glass
......

Final Answer:
The time interval required for the light to pass through the glass block is approximately:
\[t \approx 1.16 \times 10^{-10} \, \text{s}.\]

**Reasoning Type:** Physical Model Grounding Reasoning , Spatial Relation Reasoning

**Error Type:** Text Reasoning Error

**Ground Truth:** 106ps

Figure 24: A sample error case of Optics. Error category: Text Reasoning Error

# Optics ⊗

**Description:**

A narrow beam of ultrasonic waves reflects off the liver tumor illustrated in figure. The speed of the wave is 10.0% less in the liver than in the surrounding medium.

**Question:**

Determine the depth of the tumor.

**GPT-4o:**

To determine the depth of the tumor, we will use the information provided in the diagram and the relationship between the speed of the ultrasonic wave in the liver and the surrounding medium. Here's the step-by-step reasoning:

Step 1: Understand the geometry of the problem
......

Step 2: Relate the geometry to the depth
The wave travels in a triangular path. The horizontal distance between the entry and exit points is $12.0$ cm, and the wave reflects symmetrically. This means the horizontal distance from the entry point to the reflection point is $6.0$ cm.
Using trigonometry, the depth $d$ of the tumor can be related to the horizontal distance and the angle of incidence:$$\tan(50.0^\circ) = \frac{d}{6.0 \, \text{cm}}.$$
Rearranging for $d$:$$d = 6.0 \, \text{cm} \cdot \tan(50.0^\circ).$$

Step 3: Step 3: Calculate $d$
......

Step 4: Adjust for the speed difference
......

Final Answer:
The depth of the tumor is approximately:
$$\boxed{7.94 \, \text{cm}}.$$

**Reasoning Type:** Spatial Relation Reasoning , Physical Model Grounding Reasoning

**Error Type:** Lack of Knowledge

**Ground Truth:** 6.30cm

Figure 25: A sample error case of Optics. Error category: Lack of Knowledge

# Modern Physics

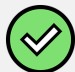

### Description:

Peggy is standing in the center of a long, flat railroad car that has firecrackers tied to both ends. The car moves past Ryan, who is standing on the ground, with velocity v = 0.8c. Flashes from the exploding firecrackers reach him simultaneously 1.0μs after the instant that Peggy passes him, and he later finds burn marks on the track 300m to either side of where he had been standing.

### Question:

According to Peggy, what times do the explosions occur relative to the time that Ryan passes her?

### GPT-4o:

To solve this problem, we will use the principles of special relativity, specifically the Lorentz transformation equations, to determine the times of the explosions in Peggy's frame ($S'$) relative to the time that Ryan passes her ($t' = 0$).

Step 1: Understand the problem and given data
Ryan's frame ($S$):
- The firecrackers explode at $x_L = -300\;m$ and $x_R = 300\;m$, both at $t = 0$.
- The flashes from the explosions reach Ryan simultaneously at $t = 1.0\;\mu s$.
- The velocity of Peggy's frame ($S'$) relative to Ryan's frame ($S$) is $v = 0.8c$.
Peggy's frame ($S'$):
- Peggy is at rest in her frame, and the firecrackers are tied to the ends of the railroad car.
- We need to find the times of the explosions ($t'_L$ and $t'_R$) in Peggy's frame.

Step 2: Lorentz transformation equations
The Lorentz transformation equations relate the coordinates $(x, t)$ in Ryan's frame ($S$) to the coordinates $(x', t')$ in Peggy's frame ($S'$): $$ t' = \gamma \left( t - \frac{vx}{c^2} \right), $$ $$ x' = \gamma \left( x - vt \right), $$
where: $$ \gamma = \frac{1}{\sqrt{1 - v^2/c^2}}. $$
For $v = 0.8c$, we calculate: $$ \gamma = \frac{1}{\sqrt{1 - (0.8)^2}} = \frac{1}{\sqrt{1 - 0.64}} = \frac{1}{\sqrt{0.36}} = \frac{5}{3}. $$

Step 3: Apply the Lorentz transformation for each event
Event L (explosion at $x_L = -300\;m$, $t_L = 0$):
Using the Lorentz transformation for time: $$ t'_L = \gamma \left( t_L - \frac{v x_L}{c^2} \right). $$
Substitute the values: $$ t'_L = \frac{5}{3} \left( 0 - \frac{(0.8c)(-300\;m)}{c^2} \right). $$
Simplify: $$ t'_L = \frac{5}{3} \left( \frac{240\;m}{c} \right). $$
Since $c = 3 \times 10^8\;m/s$, we calculate: $$ t'_L = \frac{5}{3} \cdot \frac{240}{3 \times 10^8} = \frac{5\cdot 240}{9 \times 10^8} =\frac{1200}{9 \times 10^8} = \frac{4}{3} \times 10^{-6}\;s = 1.33\;\mu s. $$
Event R (explosion at $x_R = 300\;m$, $t_R = 0$):
Using the Lorentz transformation for time: $$ t'_R = \gamma \left( t_R - \frac{v x_R}{c^2} \right). $$
Substitute the values: $$ t'_R = \frac{5}{3} \left( 0 - \frac{(0.8c)(300\;m)}{c^2} \right). $$
Simplify: $$ t'_R = \frac{5}{3} \left( -\frac{240\;m}{c} \right). $$ $$ t'_R = -\frac{5}{3} \cdot \frac{240}{3 \times 10^8} = -\frac{5 \cdot 240}{9 \times 10^8} = -\frac{1200}{9 \times 10^8} = -\frac{4}{3} \times 10^{-6}\;s = -1.33\;\mu s. $$

Step 4: Interpret the results
- In Peggy's frame ($S'$):
- The explosion at $x'_L$ (left firecracker) occurs at $t'_L = 1.33\;\mu s$.
- The explosion at $x'_R$ (right firecracker) occurs at $t'_R = -1.33\;\mu s$.

This means that, according to Peggy, the right firecracker explodes before Ryan passes her, and the left firecracker explodes after Ryan passes her.

**Reasoning Type: Physical Model Grounding Reasoning , Spatial Relation Reasoning**

**Ground Truth: -1.33μs**

Figure 26: A sample correct case of Modern Physics.

# Modern Physics

**Description:**

An enemy spaceship is moving toward your starfighter with a speed, as measured in your frame, of 0.400c. The enemy ship fires a missile toward you at a speed of 0.700c relative to the enemy ship. You measure that the enemy ship is $8.00\times10^6$ km away from you when the missile is fired.

**Question:**

If you measure that the enemy ship is $8.00\times10^6$ km away from you when the missile is fired, how much time, measured in your frame, will it take the missile to reach you?

**Options:**

A. 31.0s                 B. 30.4s                 C. 33.1s                 D. 32.8s

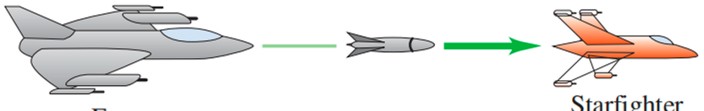

**GPT-4o:**

To solve this, we use the relativistic velocity addition formula to find the speed of the missile relative to the starfighter:
$$v_{\text{missile/starfighter}} = \frac{v_{\text{missile/enemy}} + v_{\text{enemy/starfighter}}}{1 + \frac{v_{\text{missile/enemy}} v_{\text{enemy/starfighter}}}{c^2}}$$
Substitute the given values:
$$v_{\text{missile/enemy}} = 0.700c, \quad v_{\text{enemy/starfighter}} = 0.400c$$
$$v_{\text{missile/starfighter}} = \frac{0.700c + 0.400c}{1 + \frac{(0.700c)(0.400c)}{c^2}}$$
$$v_{\text{missile/starfighter}} = \frac{1.100c}{1 + 0.280} = \frac{1.100c}{1.280} \approx 0.859c$$

Now, calculate the time it takes for the missile to travel the distance of $8.00 \times 10^6 \, \text{km}$ (or $8.00 \times 10^9 \, \text{m}$) at this speed:
$$t = \frac{\text{distance}}{\text{speed}} = \frac{8.00 \times 10^9 \, \text{m}}{0.859c}$$

Substitute $c = 3.00 \times 10^8 \, \text{m/s}$:
$$t = \frac{8.00 \times 10^9}{0.859 \times 3.00 \times 10^8} \approx \frac{8.00}{0.859 \times 3.00} \times 10^1$$
$$t \approx \frac{8.00}{2.577} \times 10^1 \approx 3.10 \times 10^1 \, \text{s} = 31.0 \, \text{s}$$

The correct answer is: A

**Reasoning Type: Physical Model Grounding Reasoning , Multi-Formula Reasoning**

**Ground Truth: A . 31.0s**

Figure 27: A sample correct case of Modern Physics.

# ModernPhysics ⊗

**Description:**

An experiment was performed in which neutrons were shot through two slits spaced 0.10 nm apart and detected 3.5m behind the slits. Figure shows the detector output.

**Question:**

To one significant figure, what was the speed of the neutrons?

**GPT-4o:**

To determine the speed of the neutrons, we can use the principles of wave interference and de Broglie's hypothesis. Here's the step-by-step reasoning:

Step 1: Understand the setup
The experiment involves neutrons passing through two slits spaced $d = 0.10 \,\text{nm} = 1.0 \times 10^{-10} \,\text{m}$ apart. The interference pattern is detected on a screen located $L = 3.5 \,\text{m}$ behind the slits. The distance between adjacent maxima in the interference pattern is $\Delta y = 100 \,\mu\text{m} = 1.0 \times 10^{-4} \,\text{m}$.

Step 2: Use the double-slit interference formula
The position of the maxima in a double-slit interference pattern is given by:
$$y_m = m \frac{\lambda L}{d},$$
...

Step 3: Relate the wavelength to the speed of the neutrons
Using de Broglie's hypothesis, the wavelength of a particle is related to its momentum by:
$$\lambda = \frac{h}{p},$$
where:
- $h = 6.63 \times 10^{-34} \,\text{J·s}$ is Planck's constant,
- $p = mv$ is the momentum of the neutron,
- $m = 1.675 \times 10^{-27} \,\text{kg}$ is the mass of a neutron,
- $v$ is the speed of the neutron.

Rearranging for $v$:
$$v = \frac{h}{m \lambda}.$$

Substitute the known values:
$$v = \frac{6.63 \times 10^{-34} \,\text{J·s}}{(1.675 \times 10^{-27} \,\text{kg}) \cdot (2.86 \times 10^{-15} \,\text{m})}.$$
$$v = 1.38 \times 10^3 \,\text{m/s}.$$

Step 4: Round to one significant figure
To one significant figure, the speed of the neutrons is:
$$v \approx 1 \times 10^3 \,\text{m/s}.$$

Final Answer:
The speed of the neutrons is approximately $\mathbf{1 \times 10^3 \,\text{m/s}}$.

**Reasoning Type: Physical Model Grounding Reasoning , Multi-Formula Reasoning**

**Error Type: Visual Reasoning Error**

**Ground Truth: 200m/S**

Figure 28: A sample error case of Modern Physics. Error category: Visual Reasoning Error

# ModernPhysics  ⊗

### Description:

Owen and Dina are at rest in frame S, which is moving with respect to frame S. They play a game of catch while Ed, at rest in frame S, watches the action as shown in figure. Owen throws the ball to Dina.

### Question:

What time interval is required for the ball to reach Dina?

### GPT-4o:

To determine the time interval required for the ball to reach Dina, we need to analyze the situation using the principles of special relativity. Here's the step-by-step reasoning:

Step 1: Understand the problem and given data**
- Frames of reference:**
- Frame $S'$: The rest frame of Owen and Dina.
- Frame $S$: The rest frame of Ed.

Given data:**
  - The relative velocity of frame $S'$ with respect to frame $S$: $v = 0.600c$.
  - The velocity of the ball relative to $S'$: $u' = 0.800c$ (from Owen to Dina).
  - The distance between Owen and Dina in $S'$: $d' = 1.80 \times 10^{12} \, \text{m}$.

We are tasked with finding the time interval for the ball to reach Dina **as observed in frame $S$**.

Step 2: Relate the velocity of the ball in $S$ to the velocity in $S'$**
The velocity of the ball in frame $S$ ($u$) can be calculated using the relativistic velocity addition formula:

$$ u = \frac{u' + v}{1 + \frac{u'v}{c^2}} $$

Substitute the given values:
- $u' = 0.800c$
- $v = 0.600c$

$$ u = \frac{0.800c + 0.600c}{1 + \frac{(0.800c)(0.600c)}{c^2}} $$

...
Step 3: Calculate the time interval in frame $S$
...

Final Answer:
The time interval required for the ball to reach Dina, as observed in frame $S$, is approximately:
$$ \boxed{6.34 \, \text{seconds}} $$

Reasoning Type: Physical Model Grounding Reasoning ,  Multi-Formula Reasoning

Error Type: Text Reasoning Errors

Ground Truth: 4.88 * 10^3 s

Figure 29: A sample error case of Modern Physics. Error category: Text Reasoning Error

# ModernPhysics

### Description:

Figure shows the wave function of an electron in a rigid box. The electron energy is 6.0eV.

$\psi(x)$

### Question:

How long is the box?

### GPT-4o:

To determine the length of the box, we will use the quantum mechanical model of a particle in a rigid box. The energy levels of an electron in a rigid box are given by:

\[ E_n = \frac{n^2 h^2}{8mL^2} \]

where:
- \( E_n \) is the energy of the electron in the \( n \)-th energy level,
- \( n \) is the quantum number (a positive integer),
- \( h \) is Planck's constant (\( h = 6.626 \times 10^{-34} \, \text{J·s} \)),
- \( m \) is the mass of the electron (\( m = 9.109 \times 10^{-31} \, \text{kg} \)),
- \( L \) is the length of the box.

Step 1: Identify the quantum number \( n \)
...
Step 4: Convert to nanometers
\[ L = 7.52 \times 10^{-10} \, \text{m} = 0.752 \, \text{nm}. \]

Final Answer:
The length of the box is approximately:
\[ \boxed{0.752 \, \text{nm}} \]

### Reasoning Type: Physical Model Grounding Reasoning , Spatial Relation Reasoning

### Error Type: Lack of Knowledge

### Ground Truth: 1 nm

Figure 30: A sample error case of Modern Physics. Error category: Lack of Knowledge

# I    DATA ANNOTATION PROTOCOL

This document outlines a detailed procedure for annotating a dataset of physics questions that include visual context.

## I.1    DATA COLLECTION

**Sources of Data.**    Data is collected from freely accessible online resources, textbooks, and other materials. Annotators are instructed to use a wide range of sources rather than relying on just one.

**Types of Questions:**

- **Multiple-Choice Questions:** These consist of a question accompanied by four answer options, with only one being correct. For each multiple-choice question, annotators are also required to create a corresponding open-ended version of the same problem.
- **Open-Ended Questions:** These include formats such as short-answer and calculation-based problems. Questions with excessively lengthy answers should be avoided. For each open-ended question, a corresponding multiple-choice version should also be constructed.

**Image Types.** The annotators should find images with realistic physical senarios.

## I.2    GENERAL GUIDELINES

- **General Principles:** Annotations should be accurate and uniform, and maintain a high level of academic quality.
- **Specific Instructions:**
  - All questions should be written in English.
  - All questions must contain one physical image.
  - All images in question should be realistic, in specific physical scenarios.
  - The question should not be ambiguous and can be answered with one of the given options or a short answer.
  - Annotate all data fields, including the description, simplified description, question, answer options, the correct answer, image, and domain.

## I.3    DATA FORMAT AND STRUCTURE

- **JSON File Format:** The structured JSON format will include fields for index number, description, simplified description, question, answer options, correct answer, and domain.
- **Naming Conventions:**
  - Each collected sample will be stored on a single line in a JSONL file.
  - Image files should follow a standard naming rule: **{QuesNum}**.png
- **Interleaving Question with Images:** The images should be inserted as a file path in the question.

## I.4    QUALITY CONTROL AND VALIDATION

- Annotators will cross-check each other's work to ensure accuracy and compliance with the annotation guidelines.
- Periodic reviews of randomly selected samples from the dataset will be carried out to maintain consistent quality over time.

## I.5    HANDLING AMBIGUITIES

Any ambiguous or unclear data entries should be marked for thorough review. Such questions will be collectively discussed during team meetings to develop a consistent and standardized annotation strategy.

### I.6 ETHICAL CONSIDERATIONS

- **Copyright and Licensing:** Annotators must strictly follow all applicable copyright and licensing rules. Content from sources that restrict reproduction or redistribution will be excluded without exception.

- **Data Privacy:** Upholding data privacy and ethical standards is essential. Annotators should refrain from including any questions that involve personal or sensitive information.

### I.7 DATA CONTAMINATION CONSIDERATIONS

When developing benchmarks for evaluating foundation models, it is crucial to account for the potential risk of data contamination. To mitigate this, annotators should deliberately avoid simple questions with widely available answers. Instead, they should prioritize selecting problems whose solutions are embedded in less conspicuous places—such as in supplementary materials or at the end of lengthy textbooks. This strategy helps ensure that the benchmark effectively challenges models to demonstrate genuine comprehension and reasoning across complex and less accessible content.

