# OpenReview forum: "PhyX: Does Your Model Have the "Wits" for Physical Reasoning?"
_ICLR.cc/2026/Conference — ICLR 2026 Conference Withdrawn Submission_

### Official Review · Reviewer_7xzb · 2025-10-29

**Soundness:** 3
**Presentation:** 3
**Contribution:** 2
**Rating:** 4
**Confidence:** 3

**Summary:**

This work introduces a new benchmark PhyX for physical reasoning with visual components of 3k problems and the results reveal that the physical understanding ability of existing models is still lacking.

**Strengths:**

- The dataset collected covers a wide range of physics problems.
- The text deredundancy step is useful in isolating the multimodal information and avoid information compensation from the text description.
- The data collection process includes several human verification steps to maintain the quality.

**Weaknesses:**

- The expert background is not well introduced. For example, whether they have different strength under the subdomains in the dataset and whether it is considered in the evaluation.
- The findings and analysis does not provide enough insights to sharpen the understanding of the boundary of existing LLMs.

**Questions:**

- The worst/medium/best performance of human expert is not explained, is this identical to min/medium/max performance among all participants, or it is related to the expert group? The context seem to indicate the latter but I am not entirely sure.
- Are the students within the same group able to communicate or collaborate to solve the problem?
- Are we supposed to transfer the performance of text deredundancy as the approximation to the full-text or text-minimal setting for human expert? Do you have any knowledge about this generalization?
- In Table 3, the strongest model is able to surpass the medium expert under some settings, can you provide more details regarding this phenomenon?
- Where are the original source of the problems, and images if introduced elsewhere?

---

### Official Review · Reviewer_cgQS · 2025-10-29

**Soundness:** 2
**Presentation:** 2
**Contribution:** 2
**Rating:** 2
**Confidence:** 5

**Summary:**

The paper introduces PHYX, a benchmark for physics grounded reasoning in multimodal settings. It offers 3k unique visual physics problems with both multiple choice and open ended formats, spans six domains and six reasoning types, and provides three text variants to probe text reliance. The authors evaluate many recent LLMs and MLLMs and present error analyses and takeaways about visual grounding and modality fusion. The topic is timely and the dataset could be useful to the community.

**Strengths:**

- Important problem focus. Physical reasoning that integrates perception, symbolic manipulation, and real world constraints is a valuable target for evaluation.
- Breadth of coverage across six physics domains and six reasoning categories with both MC and OE formats.
- Three input variants to study redundancy and text dependence.
- Integration with common eval toolkits and release plan for one click evaluation.
- Evaluation includes both MLLMs and text only LLMs through captions, which enables cross modality comparisons.

My recommendation: reject.
1. Methodological weaknesses in evaluation and judging reduce confidence in the reported gaps and rankings.
2. Incremental novelty relative to recent physics reasoning benchmarks, with several claims that appear overstated.

**Weaknesses:**

- Novelty claim appears overstated. Several 2025 benchmarks already target physics reasoning with images, for example PhysReason, UGPhysics, SeePhys, PhysUniBench, and others. PHYX is larger and uses some forms of de redundancy, but the claim of first large scale benchmark is not well supported.
- Human baseline is too small and too weak (table 2 is basically empty for the human baselines). Only 15 students answered 18 questions each, with no per question overlap, no variance estimates, and seemingly only in one setting. This cannot support strong claims about a persistent 10 point human to model gap.
- Moreover, MC accuracy for GPT-5 reaches 90.9! in test which conflicts with the headline message that "all models struggle", yet this is not reconciled with the human comparison.
- Dataset scale and counts are confusing. The paper alternates between 3k and 6k questions. Table 1 says total new questions 6k yet unique questions 3k, and the text claims 3k problems.
- Claims of realistic images (a central assertion in the paper, as shown in Fig. 3) are inconsistent with numerous examples that resemble textbook-style drawings. The paper itself later clarifies that these are not photographs. Therefore, the claim that PHYX uses realistic scenes should be moderated or better substantiated. Statements such as “We observe that the images in our dataset are highly realistic, often depicting concrete physical scenarios rather than stylized or abstract illustrations” must be supported, especially when they form part of the paper’s main claims.
- Evaluation details are under specified. CoT prompting, temperatures, seeds, and retries are not fixed.
- Filtering out the shortest 10 percent of questions as a quality control step is a weak proxy for difficulty. This is definitely not a "rigorous process (that) plays a crucial role in maintaining the quality and difficulty of PHYX". In reality, short questions can be the hardest ones!
- The paper is generally difficult to read, with long sentences containing many redundant words and unnecessary phrases that do not contribute to the content and serve only as fillers.

**Questions:**

- How was the de redundancy edit validated. Did independent annotators confirm that answers remain unchanged and that difficulty is comparable?
- See the weaknesses.

---

### Official Review · Reviewer_i2yj · 2025-10-31

**Soundness:** 3
**Presentation:** 3
**Contribution:** 2
**Rating:** 4
**Confidence:** 4

**Summary:**

The paper introduces PhyX, a benchmark designed to evaluate the physical-reasoning capabilities of MLLMs and LLMs. Itcovers six core physics domains and includes around 3 000 multimodal questions across 6 reasoning types and 25 sub-domains. Experiments show that current strong models achieve only modest accuracy on PhyX, showing a significant gap to humans. The paper also offers fine‐grained analysis from diverse perspectives, which leads to several findings.

**Strengths:**

1. The proposed physical reasoning tasks are indeed important and crucial to model intelligence.

2. The construction process of the benchmark is detailed and reasonable.

3. Experiments are conducted comprehensively, which leads to several findings.

**Weaknesses:**

1. From my understanding, the authors try to equate “physical reasoning” with the ability to solve challenging physics problems, suggesting that a model performing well on such tasks demonstrates strong reasoning capabilities. However, if a model performs well on gravity-related problems, can it generalize the same underlying principles to buoyancy, which essentially involves the same concept but in the opposite force direction? This kind of "generalization" or "learning the core idea" are also important aspects of physical reasoning.

2. It will be interesting if the authors can compare PhyX with [1], which investigates "physical reasoning" from an abstract input format perspective.

3. In a benchmarking paper, while I admit that finding problems is important, it may be good to see some potential ways to mitigate the found issues based on the findings.

[1] Yu M, Liu L, Wu J, et al. The Stochastic Parrot on LLM's Shoulder: A Summative Assessment of Physical Concept Understanding[J]. arXiv preprint arXiv:2502.08946, 2025.

**Questions:**

1. Could the authors provide a clear definition of "physical reasoning" under the scope of this paper?

2. Could you provide some potential ways (with some results) to mitigate the found issues?

---

### Note · Authors · 2025-11-12

I have read and agree with the venue's withdrawal policy on behalf of myself and my co-authors.